# COPER: CORRELATION-BASED PERMUTATIONS FOR MULTI-VIEW CLUSTERING

**Ran Eisenberg**[*], **Jonathan Svirsky**[*], **Ofir Lindenbaum**
Faculty of Engineering
Bar Ilan University
Ramat Gan, 5290002, Israel
`{ran.eisenberg,jonathan.svirsky,ofir.lindenbaum}@biu.ac.il`

## ABSTRACT

Combining data from different sources can improve data analysis tasks such as clustering. However, most of the current multi-view clustering methods are limited to specific domains or rely on a suboptimal and computationally intensive two-stage process of representation learning and clustering. We propose an end-to-end deep learning-based multi-view clustering framework for general data types (such as images and tables). Our approach involves generating meaningful fused representations using a novel permutation-based canonical correlation objective. We provide a theoretical analysis showing how the learned embeddings approximate those obtained by supervised linear discriminant analysis (LDA). Cluster assignments are learned by identifying consistent pseudo-labels across multiple views. Additionally, we establish a theoretical bound on the error caused by incorrect pseudo-labels in the unsupervised representations compared to LDA. Extensive experiments on ten multi-view clustering benchmark datasets provide empirical evidence for the effectiveness of the proposed model.

## 1 INTRODUCTION

Clustering is an important task in data-driven scientific discovery that focuses on categorizing samples into groups based on semantic relationships. This allows for a deeper understanding of complex datasets and is used in various domains such as gene expression analysis in bioinformatics (Armingol et al., 2021), efficient categorizing of large-scale medical images (Kart et al., 2021), and in collider physics (Mikuni & Canelli, 2021). Existing clustering approaches can be broadly categorized as centroid-based (Boley et al., 1999; Jain, 2010; Velmurugan & Santhanam, 2011), density-based (Januzaj et al., 2004; Kriegel & Pfeifle, 2005; Chen & Tu, 2007; Duan et al., 2007), distribution-based (Preheim et al., 2013; Jiang et al., 2011a), and hierarchical (Murtagh, 1983; Carlsson et al., 2010). Multi-view clustering is an extension of the clustering paradigm that simultaneously leverages diverse views of the same observations (Sun & Tao, 2014; Kumar et al., 2011; Li et al., 2018b).

The main idea behind multi-view clustering is to combine information from multiple data facets (or views) to obtain a more comprehensive and accurate understanding of the underlying data structures. Each view may capture distinct aspects or facets of the data, and by integrating them, we can discover hidden patterns and relationships that might be obscured in any single view (Huang et al., 2012; Xu et al., 2013). This approach has great potential in applications like communication systems content delivery (Vázquez & Pérez-Neira, 2020), community detection in social networks (Zhao et al., 2022; Shi et al., 2022), cancer subtype identification in bioinformatics (Wen et al., 2021), and personalized genetic analysis through multi-modal clustering frameworks (Li et al., 2023; Kuang et al., 2024).

Existing multiview clustering (MVC) methods can be divided into traditional (non-deep) and deep learning-based methods. Traditional MVC methods include: subspace (Cao et al., 2015; Luo et al., 2018; Li et al., 2019a), matrix factorization (Zhao et al., 2017; Wen et al., 2018; Yang et al., 2021), and graph learning-based methods (Nie et al., 2017; Zhan et al., 2017; Zhu et al., 2018). The main drawbacks of the traditional methods are poor representation ability, high computation complexity, and limited performance in real-world data (Guo & Ye, 2019).

---

[*]Equal contribution

Recently, several deep learning-based MVC schemes have demonstrated promising representation and clustering capabilities (Abavisani & Patel, 2018; Alwassel et al., 2019; Li et al., 2019b; Yin et al., 2020; Wen et al., 2020; Xu et al., 2021a;b). Most of these methods adopt a two-stage approach, where they first learn representations, followed by clustering, as seen in works such as Lin et al. (2021); Xu et al. (2021b); Abavisani & Patel (2018); Alwassel et al. (2019); Li et al. (2019b); Cao et al. (2015); Zhang et al. (2017a); Brbić & Kopriva (2018); Tang et al. (2018). However, such a two-stage procedure can be computationally expensive and does not directly update the model's weights based on cluster assignments; therefore, it may lead to suboptimal results.

A few studies presented an end-to-end scheme for MV representation learning and clustering (Tang & Liu, 2022; Trosten et al., 2021b; Chen et al., 2020; 2023; Sun et al., 2024; Chao et al., 2024; Gupta et al., 2024)). By performing both tasks simultaneously, MVC can improve the data embeddings by making them more suited for cluster assignments. However, the multi-view fusion process used in these studies may not be adaptable to all types of data, which can limit their generalization capabilities across a wide range of datasets.

We introduce a new approach called COrrelation-based PERmutations (COPER) that aims to address the main challenges of multi-view clustering. Our deep learning model combines clustering and representation learning tasks, providing an end-to-end MVC framework for fusion and clustering. This eliminates the need for an additional step. The approach involves learning data representations through a novel self-supervision technique, where within-class pseudo-labels are permuted (PER) across different views for canonical correlation (CO) analysis loss. The proposed framework approximates the same projection that would have been achieved by the (supervised) linear discriminant analysis (LDA) method (Fisher, 1936), under some mild assumptions. This projection enhances clustering capabilities as it maximizes between-class variance while minimizing within-class variation.

Our main contributions are summarized as follows: (i) Develop a deep learning model that exploits self-supervision and a CCA-based objective for end-to-end MVC. (ii) Present a multi-view pseudo-labeling procedure for identifying consistent labels across views. (iii) Demonstrate empirically and theoretically that within-cluster permutation can improve the usability of CCA-based representations for MVC by enhancing cluster separation. (iv) Analyze the relation between the solution of our new permutation-based CCA procedure and the solution obtained by the supervised LDA. Which further justifies the

Table 1: Average performance.

| Method | ACC | ARI | NMI |
|---|---|---|---|
| DSMVC | 48.6 | 34.1 | 47.2 |
| CVCL | 51.1 | 37.1 | 48.5 |
| ICMVC | 45.5 | 30.3 | 41.6 |
| RMCNC | 40.7 | 22.1 | 31.0 |
| OPMC | 55.9 | 41.4 | 52.9 |
| MVCAN | 44.1 | 29.5 | 40.9 |
| AE | 38.6 | 19.4 | 30.9 |
| DCCA-AE | 47.8 | 29.3 | 41.4 |
| $\ell_0$-DCCA | 44.2 | 26.8 | 40.8 |
| **COPER** | **61.6** | **47.8** | **54.0** |

usability of our model for MVC. This analysis indicates that our method can be applied to a broader range of CCA-based methods. (v) Conduct a comprehensive experimental evaluation to demonstrate the superiority of our proposed model over the state-of-the-art deep MVC models. In Table 1, we summarize the average performance for all baselines on ten datasets: COPER improves the best baselines in ACC, ARI, and NMI metrics.

## 2 BACKGROUND

### 2.1 CANONICAL CORRELATION ANALYSIS (CCA)

Canonical Correlation Analysis (CCA) (Harold, 1936; Thompson, 1984) is a well-celebrated statistical framework for multi-view/modal representation learning. CCA can help analyze the associations between two sets of paired observations. This framework and its nonlinear extensions (Bach & Jordan, 2002; Michaeli et al., 2016; Lindenbaum et al., 2020; Salhov et al., 2020; Andrew et al., 2013) have been applied in various domains, including biology (Pimentel et al., 2018), neuroscience (Al-Shargie et al., 2017), medicine (Zhang et al., 2017b), and engineering (Chen et al., 2017).

The main goal of CCA is to find linear combinations of variables from each view, aiming to maximize their correlation. Formally, denoting the observations as $\boldsymbol{X}^{(1)} \in \mathbb{R}^{D_1 \times N}$ and $\boldsymbol{X}^{(2)} \in \mathbb{R}^{D_2 \times N}$, where both modalities are centered and encompass $N$ samples with $D_1$ and $D_2$ attributes, respectively. CCA seeks for *canonical vectors* $\boldsymbol{a} \in \mathbb{R}^{D_1}$ and $\boldsymbol{b} \in \mathbb{R}^{D_2}$ such that $\boldsymbol{u} = \boldsymbol{a}^T \boldsymbol{X}^{(1)}$ and $\boldsymbol{v} = \boldsymbol{b}^T \boldsymbol{X}^{(2)}$. The objective is to maximize correlations between these canonical variates, as represented by the

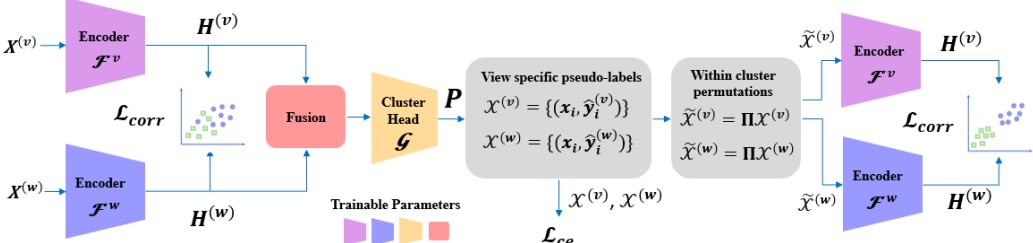

Figure 1: Our proposed deep learning model, COrrelation-based PERmutations (COPER). Two modalities $v$ and $w$, are processed through view-specific encoders, creating latent embeddings ($\boldsymbol{H}^{(v)}$ and $\boldsymbol{H}^{(w)}$). The embeddings are learned using a correlation maximization loss and then fused to serve as input for the clustering head. The clustering head estimates a probability matrix, denoted as $P$, which is used to derive multi-view pseudo-labels. We then generate within-cluster permutations based on these pseudo-labels. The permuted samples are used to update the CCA representation. By changing the pairing of observations fed into the CCA objective, we enhance cluster separation and extract embeddings that theoretically approximate the solution of supervised Linear Discriminant Analysis (LDA), as demonstrated in Section 4.3.

following optimization task:

$$\max_{\boldsymbol{a},\boldsymbol{b}\neq 0} \frac{\boldsymbol{a}^T\boldsymbol{X}^{(1)}(\boldsymbol{X}^{(2)})^T\boldsymbol{b}}{\sqrt{\boldsymbol{a}^T\boldsymbol{X}^{(1)}(\boldsymbol{X}^{(1)})^T\boldsymbol{a}}\sqrt{\boldsymbol{b}^T\boldsymbol{X}^{(2)}(\boldsymbol{X}^{(2)})^T\boldsymbol{b}}}. \tag{1}$$

These canonical vectors can be found by solving the generalized eigenpair problem

$$C_1^{-1}C_{12}C_2^{-1}C_{21}\boldsymbol{a} = \lambda\boldsymbol{a}, \quad C_2^{-1}C_{21}C_1^{-1}C_{12}\boldsymbol{b} = \lambda\boldsymbol{b},$$

where $C_1, C_2$ are within view sample covariance matrices and $C_{12}, C_{21}$ are cross-view sample covariance matrices.

Various extensions of CCA have been proposed to study non-linear relationships between the observed modalities. Some kernel-based methods, such as Kernel CCA (Bach & Jordan, 2002), Non-parametric CCA (Michaeli et al., 2016), and Multi-view Diffusion maps (Lindenbaum et al., 2015; Salhov et al., 2020), explore non-linear connections within reproducing Hilbert spaces. However, these methods are limited by pre-defined kernels, have restricted interpolation capabilities, and do not scale well with large datasets.

To overcome these limitations Andrew et al. (2013) introduced Deep CCA (DCCA), which extends traditional CCA by leveraging neural networks to model non-linear interactions between input features. This enables more flexible and scalable modeling of complex relationships in large datasets. Section 4.4.1 describes how we incorporate a DCCA objective to embed the multi-view data.

## 2.2 Linear Discriminant Analysis (LDA)

Fisher's linear discriminant analysis (LDA) aims to preserve variance while seeking the optimal linear discriminant function (Fisher, 1936). Unlike unsupervised techniques such as principal component analysis (PCA) or canonical correlation analysis (CCA), LDA is a supervised method that incorporates categorical class label information to identify meaningful projections. LDA's objective function is designed to be maximized through a projection that increases the between-class scatter and reduces the within-class scatter.

For a dataset $\boldsymbol{X} \in \mathbb{R}^{N \times D}$ and it's covariance matrix $C$, we denote the within-class covariance matrix as $C_e$ and the between-cluster covariance matrix as $C_a$.

The optimization for LDA can be formulated as the following form (left), and its corresponding generalized eigenproblem (right):

$$\max_{\boldsymbol{h}\neq 0} \frac{\boldsymbol{h}^T C_a \boldsymbol{h}}{\boldsymbol{h}^T C_e \boldsymbol{h}}; \quad \rightarrow \quad C_a \boldsymbol{h} = \lambda C_e \boldsymbol{h}, \quad C_e^{-1}C_a \boldsymbol{h} = \lambda \boldsymbol{h}, \tag{2}$$

where $h$ is the eigenvector that maximizes the LDA objective. The representation obtained by $h$ is ideal for clustering as it aims to maximize the distance between class means while minimizing the variance within each class. A detailed formulation of LDA can be found in Appendix G.1. Despite the ideal properties of LDAs representation, it is not traditionally used for unsupervised learning tasks such as clustering as it requires labeled data for its objective. Several existing works use LDA adaptations for feature extraction and dimensionality reduction for downstream clustering (Pang et al., 2014; Zhu & Hastie, 2003).

### 2.3 SELF-SUPERVISION FOR CLUSTERING

Self-supervised learning is a method for acquiring meaningful data representations without using labeled data. It leverages information from unlabeled samples to create tasks that do not require manual annotations. In clustering tasks, self-supervision improves data representation learning by assigning pseudo-labels to unlabeled data based on semantic similarities between samples.

In single-view data, two-stage clustering frameworks proposed by Caron et al. (2018; 2020); Nousi & Tefas (2020) alternate between clustering and using the cluster assignments as pseudo-labels to revise image representations. Niu et al. (2022) have introduced a pseudo-labeling method that encourages the formation of more meaningful and coherent clusters that align with the semantic content of the images. Their framework treats some pseudo-labels as reliable, synergizing the similarity and discrepancy of the samples. For multi-view datasets, clustering and self-supervision frameworks (Xin et al., 2021; Alwassel et al., 2019; Li et al., 2018a) offer loss components that enforce consensus in the latent multi-view representations. We propose a novel pseudo-labeling scheme suited for a multi-view setting, this scheme is suited for general data (image, tabular, etc.) and does not required any prior, domain specific knowledge. One of our method's critical and novel components is a simple update of the corresponding pairing of samples in multi-view data. Furthermore, we show a theoretical analysis that the obtained, updated CCA representation enhances cluster separation and is, therefore, suited for clustering.

## 3 RELATED WORK

Several existing MVC methods incorporate Deep Canonical correlation analysis (DCCA) during their representation learning phase. These methods obtain a useful representation by transforming multiple views into maximally correlated embeddings using nonlinear transformations (Xin et al., 2021; Chandar et al., 2016; Cao et al., 2015; Tang et al., 2018). However, these methods use a suboptimal two-stage procedure, first creating the representation learned through DCCA and then independently applying the clustering scheme.

A few end-to-end, multi-view DCCA-based clustering solutions have been proposed; these include Tang et al. (2018), which update their representations to improve the clustering capabilities. We have developed a model that belongs to the category of end-to-end representation learning and clustering. However, we have introduced several new elements that set our approach apart from existing schemes. These include a novel self-supervised permutation procedure that enhances the representation, as well as a multi-view pseudo-label selection scheme. Empirical results show that these new components significantly improve clustering capabilities, as presented in Section 5.1. In addition, our self-supervised permutation procedure is invariant to any CCA-based method.

## 4 THE PROPOSED METHOD

### 4.1 PROBLEM SETUP

We are given a set of multi-view data $\mathcal{X} = \left\{ \boldsymbol{X}^{(v)} \in \mathbb{R}^{d_v \times N} \right\}_{v=1}^{n_v}$ with $n_v$ views and $N$ samples. Each view is defined by $\boldsymbol{X}^{(v)} = [\boldsymbol{x}_1^{(v)}, \boldsymbol{x}_2^{(v)}, ..., \boldsymbol{x}_N^{(v)}]$ and each $\boldsymbol{x}_i^{(v)}$ is a $d_v$-dimensional instance. Our objective is to predict cluster assignment $y_i$ for each tuple of instances $(\boldsymbol{x}_i^{(1)}, \boldsymbol{x}_i^{(2)}, ..., \boldsymbol{x}_i^{(n_v)})$, $i = 1, ..., N$ in $\mathcal{X}$ and the number of clusters is $K$.

## 4.2 HIGH LEVEL SOLUTION

At the core of our proposed solution are three complementary components based on multi-view observations: (i) end-to-end representation learning, (ii) multi-view reliable pseudo-labels prediction, and (iii) within-cluster sample permutations.

Our representation learning component aims to create aligned data embedding by using a maximum correlation objective. This embedding captures the shared information across multiple views of the data. More details about this component can be found in Section 4.4.1. The second component involves fusing the embeddings and predicting pseudo labels using a clustering head. We then present a filtration procedure to select reliable representatives in each cluster. Both pseudo and reliable labels are used for self-supervision as they enforce agreement between the embeddings of different views. This component is described in Section 4.4.2.

Our final component uses the selected reliable label representatives and introduces permutation to the samples belonging to the same pseudo-labels. The permuted samples are then used to update the CCA representation, approximating LDA and thereby improving clustering performance (see Section 4.3). Through both theoretical and empirical analysis, we show that these permutations can enhance cluster separation when using CCA-based objectives. An overview of our model, called COPER, is presented in Fig. 1, and a complete description of the method appears in Section 4.4. We now introduce the theoretical justification for the within-cluster permutations, which are a core component of our method.

## 4.3 THEORETICAL GUARANTEES

COPER incorporates within-cluster permutation to enhance the learned multi-view embedding. The justification for this technique lies in two theoretical aspects. The first follows Kursun et al. (2011), which artificially created multi-view data from a single view by pairing samples with other samples from the same class. The authors then demonstrate that applying CCA to such data is equivalent to linear discriminant analysis (LDA) (Fisher, 1936) (described in 2.2). Here, as part of our contribution, we show theoretically that within-cluster permutations of multi-view data lead to similar results (see Section 4.3.1 and 4.3.2). This is also backed up by empirical evidence presented in Figure 3.

The second theoretical justification for our permutations was presented by Chaudhuri et al. (2009). The authors showed that CCA-based objectives can improve cluster separation if the views are uncorrelated for samples within a given cluster. Our within-cluster permutations help obtain this property, as we demonstrate empirically in Figure 7 in Appendix F.

**Definition 4.1** *A permutation is defined for the vector of indices $\bar{I}_k \subseteq 1, ..., \bar{N}_k$. These are indices of $N_k$ samples whose pseudo-labels correspond to cluster $k$. The within-cluster permutation is defined as the application of the following operator $\Pi_k^l \bar{I}_k$, where $\Pi_k^l$ is randomly sampled from the set of all permutation matrices of size $\bar{N}_k \times \bar{N}_k$. A permutation for all clusters is denoted as $\Pi^l$.*

Where $l$ represents the permutation index, as permutations may be applied multiple times across different training batches and epochs (refer to Figure 2). This flexibility allows our model to adapt to various scenarios. Once these permutations are applied to samples across the different views ($v = 1, ..., n_v$), we can augment the data with new artificially paired samples across views. In the following subsection, we show a connection between the solution of CCA induced by this within-cluster permutation augmentation procedure and the solution of LDA.

### 4.3.1 APPROXIMATION OF LDA

In Kursun et al. (2011), the authors constructed an artificial multi-view dataset from a single view by pairing samples with other samples from the same class labels. This procedure creates a multi-view dataset in which the shared information across views is the class label. They proved that applying CCA to such augmented data converges to the solution of LDA (described in Section 2.2).

We assume all views capture a common source, a shared underlying variable denoted as $\boldsymbol{\theta}$. For instance, $\boldsymbol{\theta}$ can describe a mixture model that captures the true cluster structure in the data. Similar assumptions are also present in Benton et al. (2017); Lyu & Fu (2020). Based on this, we can prove a relation between our scheme and LDA. Formally, we start from the following assumption:

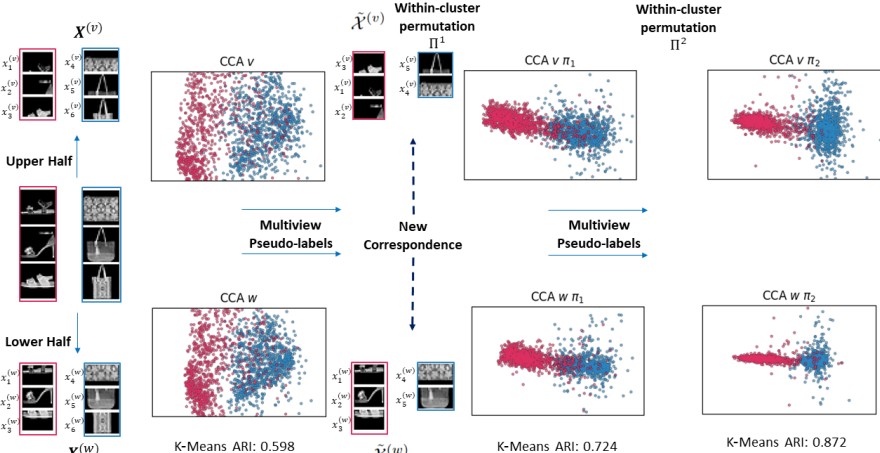

Figure 2: Illustration of how within-cluster permutations can enhance the embeddings learned by CCA. We use a binary subset of FashionMNIST and split the images to create the two views. Next, we embed the data by applying CCA to both views, $\boldsymbol{X}^{(v)}$ and $\boldsymbol{X}^{(w)}$. As described in Subsection 4.4.2, the embeddings are used to extract multi-view pseudo-labels, then within-cluster permutations $\Pi^1$ are used to create new corresponding pairs of samples $\tilde{\mathcal{X}}^{(v)}$ and $\tilde{\mathcal{X}}^{(w)}$. They are then used as augmentations to perform a second CCA (middle pair of images). This process is repeated with $\Pi^2$ (right-most pair of images). As shown by this example, using within-cluster permutations enhance the representations learned by CCA, improving clustering performance from an adjusted Rand index (ARI) of $0.598$ to $0.872$.

**Assumption 4.2** *For two different views $v, w$, the observations are created by some pushforward function (with noise) of some latent common parameter $\boldsymbol{\theta}$ that is shared across the views. Specifically, $\boldsymbol{X}^v = \boldsymbol{f}^v(\boldsymbol{\theta}, \boldsymbol{\epsilon}^v)$ and $\boldsymbol{\epsilon}^v$ is view specific noise.*

Following our assumption, $\boldsymbol{\theta}$ is shared across all views and contains the cluster information. Therefore, applying within-cluster permutations on $v, w$ (assuming there are no false cluster assignments) would be equivalent to constructing an artificial multi-view dataset in Kursun et al. (2011). We show that the following proposition holds:

**Proposition 4.3** *The embedding learned through the CCA objective using within-cluster permutation for $v$ and $w$ converges to the same representation extracted when applying the LDA objective from 2 to $\boldsymbol{\theta}$.*

A proof of this proposition appears in Appendix G.2, where we show that $\boldsymbol{h}_{LDA} = \boldsymbol{h}_{CCA}$. The proof follows the analysis of Kursun et al. (2011). Intuitively, this result indicates that the label information is "leaked" into embedding learned by CCA once we apply the within-cluster permutations. This enhances cluster separation in the obtained representation and is, therefore, suitable for clustering. In Section 4.3.3, we conduct an experiment using Fashion MNIST to demonstrate how within-cluster permutation can enhance the embedding learned by CCA. We visually illustrate how permutations can improve cluster embeddings in Figure 2.

### 4.3.2 ERROR BOUND

Since our model relies on pseudo-labels, these can induce errors in the within-cluster permutation, influencing the learned representations. If such label errors are induced, Proposition 4.3 breaks, and the solution of CCA with permutations is no longer equivalent to the solution of LDA.

To quantify this effect, we treat these induced errors as a perturbation matrix. We substitute $\boldsymbol{C}_e^{-1}\boldsymbol{C}_a$ from Eq. 2 with $\boldsymbol{A}$ and denote $\boldsymbol{D}$ as the perturbation noise such that $\hat{\boldsymbol{A}} = \boldsymbol{A} + \boldsymbol{D}$. In addition, we use tools from perturbation theory (Stewart & Sun, 1990) to provide the following upper bound for the approximated LDA eigenvalues: $|\hat{\lambda}_i - \lambda_i| \leq ||\boldsymbol{D}||_2, i = 1 \ldots n$, where $\lambda_i$ is the $i$'th LDA eigenvalue

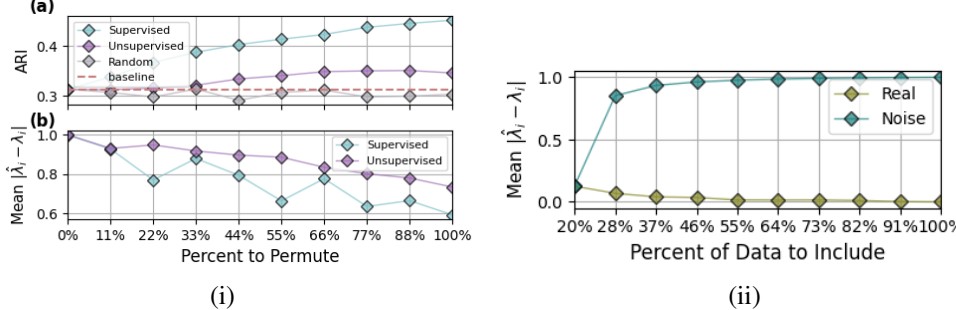

Figure 3: (i) Case study of permutation CCA using Fashion MNIST. (a) Permuting more samples within a cluster improves cluster separation as measured by the Adjusted Rand Index (ARI). We compare labeled permutation (supervised) to pseudo-label-based (unsupervised) and random. (b) Permuting more samples also pushes the representation obtained by CCA towards LDA, as indicated by the gap between eigenvalues. (ii) An experiment on LDA approximation with induced label noise. We perform LDA on different subsets of F-MNIST. We gradually increased the number of samples starting from 20% and analyzed the effect on the resulting eigenvalues $\hat{\lambda}_i$, compared to the eigenvalues obtained from LDA on the entire dataset $\lambda_i$. As expected adding more correctly annotated samples reduces the eigenvalue gap, while noisy annotations increase it.

obtained from $A$; $\hat{\lambda}_i$ is the $i$'th LDA approximated eigenvalue obtained from $\hat{A}$; and $n$ is the total number of eigenvalues. In Appendix G.3, we provide a complete derivation of this approximation. In subsection 4.3.3, we conduct a controlled experiment to evaluate how label noise influences this approximation. This implies that the more the pseudo-labels resemble the ground truth labels, the closer the representation is to LDA.

### 4.3.3 CASE STUDY ON FASHION MNIST

We conduct a controlled experiment using F-MNIST (Xiao et al., 2017) to corroborate our theoretical results presented in this section, as shown in Figure 2. First, we create two coupled views by horizontally splitting the images. CCA is subsequently performed on the multi-view dataset, with different versions of within-cluster sample permutations.

First, we only permute samples with the same label. We show in Fig. 3 (a) that such supervised permutations improve cluster separation, as evidenced by the increased ARI. Moreover, our experiments in panel (b) indicate that the CCA solution becomes more similar to the LDA solution.

Next, we repeated the evaluations in panels (a) and (b) using permutations based on pseudo-labels. We also included the results based on random permutations and the original data (no permutation) as baselines. The results show that pseudo-label-based permutations also improve cluster separation and bring the representation of CCA closer to the solution of LDA.

We conducted an additional experiment to measure the error induced by falsely annotated pseudo-labels. This experiment complements our LDA approximation presented in subsection 4.3.2. We perform LDA on different subsets of F-MNIST. We gradually increased the number of samples starting from 20% and analyzed the effect on the resulting eigenvalues, compared to the eigenvalues obtained from LDA on the entire dataset. The results, as shown in Fig. 3, demonstrate that introducing false annotations increases the gap between the eigenvalues. On the other hand, including samples with accurate annotations rapidly converges the solution to the precise LDA solution.

### 4.4 METHOD DETAILS

### 4.4.1 DEEP CANONICALLY CORRELATED ENCODERS

To learn meaningful representations from the multi-view observations, we use encoders with a maximum correlation objective (Andrew et al., 2013). Specifically, we train view-specific encoders $\mathcal{F}^{(v)}$ which extract latent representations $h_i^{(v)}$.

We utilize a correlation loss to encourage the embeddings to be correlated (see Subsection 2.1). This enables our model to extract shared information across views and reduce view-specific noise, proving

useful for multi-view clustering (Xin et al., 2021; Chandar et al., 2016; Cao et al., 2015; Tang et al., 2018).

For two different views, $v, w$ we denote the latent representations by $\boldsymbol{H}^{(v)} \in \mathbb{R}^{N \times d_v}$, and $\boldsymbol{H}^{(w)} \in \mathbb{R}^{N \times d_w}$ generated by the encoders $\mathcal{F}^{(v)}$ and $\mathcal{F}^{(w)}$ respectively. $\bar{\boldsymbol{H}}^{(v)}$ and $\bar{\boldsymbol{H}}^{(w)}$ are denoted as the centered representations of $\boldsymbol{H}^{(v)}$ and $\boldsymbol{H}^{(w)}$ respectively. The covariance matrix between these representations can be expressed as $\boldsymbol{C}_{vw} = \bar{\boldsymbol{H}}^{(v)}(\bar{\boldsymbol{H}}^{(w)})^T/(N-1)$. Similarly, the covariance matrices of $\bar{\boldsymbol{H}}^{(v)}$ and $\bar{\boldsymbol{H}}^{(w)}$ as $\boldsymbol{C}_v = \bar{\boldsymbol{H}}^{(v)}(\bar{\boldsymbol{H}}^{(v)})^T/(N-1)$ and $\boldsymbol{C}_w = \bar{\boldsymbol{H}}^{(w)}(\bar{\boldsymbol{H}}^{(w)})^T/(N-1)$. The correlation loss is defined by Eq. 3:

$$\mathcal{L}_{\mathrm{corr}}(\bar{\boldsymbol{H}}^{(v)}, \bar{\boldsymbol{H}}^{(w)}) = -\operatorname{Trace}\left[\boldsymbol{C}_w^{-1/2}\boldsymbol{C}_{wv}\boldsymbol{C}_v^{-1}\boldsymbol{C}_{vw}\boldsymbol{C}_w^{-1/2}\right]. \tag{3}$$

### 4.4.2 MULTI-VIEW PSEUDO-LABELING

A key aspect of our model is the generation of pseudo-labels, which are utilized for self-supervision to improve the representation learned by our model. Below are the main steps for creating pseudo-labels in a multi-view setting, illustrated in Figure 4. We present them concisely and provide additional details and a detailed example of these steps in Appendix D.

**Label Prediction**    The cluster head $\mathcal{G}$ predicts cluster assignments by fusing latent embeddings from multiple views $\sum_v w_v \boldsymbol{H}^{(v)}$, where $w_v$ are learnable weights. The output $\boldsymbol{P} \in \mathbb{R}^{N \times K}$ is a probability matrix where each row corresponds to the cluster probabilities for a sample. For each cluster $k$, we select the top $B$ samples with the highest probabilities to form the set $\mathcal{T}_k$ of confidently labeled samples. The union of all such sets is denoted as $\mathcal{T}$.

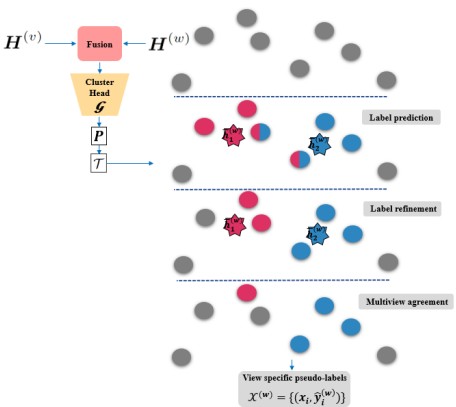

**Label Refinement**    Since samples assigned to the same cluster may have different representations across views, we refine the cluster assignments for each view separately. We compute view-specific cluster centers $\bar{\boldsymbol{h}}_k^{(v)}$ and evaluate the similarity of each sample to these centers using cosine similarity. Samples with similarity above a threshold $\lambda$ are retained, and pseudo-labels $\hat{\boldsymbol{y}}_i^{(v)}$ are created based on their similarity scores. This ensures that each sample is more accurately assigned to clusters in each view. As part of our experiment's evaluation, a detailed analysis of $\lambda$ is described in Section E.3.

Figure 4: A high-level illustration of our pseudo-labeling scheme for a single view $w$. Appendix D.1 provides a complimentary Figure (5) for the entire process, with corresponding samples in view $v$ and additional details.

**Multi-view Agreement**    For samples assigned to clusters in multiple views, we enforce agreement between views by retaining only those where the cluster predictions are consistent across views (i.e., $\operatorname{argmax}(\boldsymbol{p}_i^{(v)})$ is the same for all views).

**View-Specific Probabilities**    To train the model, view-specific probability matrices $\boldsymbol{P}^{(v)}$ are computed using $\boldsymbol{H}^{(v)}$. The remaining samples in $\mathcal{T}$ and their pseudo-labels $\hat{\boldsymbol{y}}_i^{(v)}$ are used for optimizing the cluster assignments. The final training set for each view $v$ is $\mathcal{X}^{(v)} = \{(\boldsymbol{x}_i, \hat{\boldsymbol{y}}_i^{(v)}) \mid i \in \mathcal{T}\}$.

## 5 EXPERIMENTS

### 5.1 DEEP BASELINES COMPARISON

We conduct extensive experiments with ten publicly available multi-view datasets used in recent works (Chen et al., 2023; Tang & Liu, 2022; Chao et al., 2024; Sun et al., 2024). The properties

Table 2: Clustering ACC evaluation using ten datasets. Our model (COPER) is compared against four recent end-to-end MVC models (DSMVC (Tang & Liu, 2022), CVCL (Chen et al., 2023), ICMVC (Chao et al., 2024), RMCNC (Sun et al., 2024), OPMC (Liu et al., 2021), MVCAN (Xu et al., 2024)), and two two-stage schemes. The ARI, and NMI evaluations are in Table 6

| Method | METABRIC | Reuters | Caltech101-20 | VOC | Caltech5V-7 | RBGD | MNIST-USPS | CCV | MSRVC1 | Scene15 |
|---|---|---|---|---|---|---|---|---|---|---|
| | | | | | ACC | | | | | |
| DSMVC | 40.60±3.8 | 46.37±4.4 | 39.33±2.4 | 57.82±5.0 | 79.24±9.5 | 39.77±3.6 | 70.06±10.3 | 17.90±1.2 | 60.71±15.2 | 34.30±2.9 |
| CVCL | 42.66±6.2 | 45.06±8.0 | 33.50±1.4 | 36.88±3.1 | 78.58±5.0 | 31.04±1.8 | 99.38±0.1 | 26.23±1.9 | 77.90±12.3 | 40.16±1.8 |
| ICMVC | 32.12±1.16 | 38.18±0.78 | 26.54±0.52 | 36.94±1.29 | 60.97±2.37 | 32.97±1.23 | 99.29±0.08 | 20.88±1.21 | 66.28±6.11 | **41.26±0.89** |
| RMCNC | 32.55±1.30 | 37.61±1.50 | 36.73±1.38 | 39.66±1.48 | 56.77±2.98 | 33.22±1.85 | 76.37±4.13 | 21.46±0.69 | 32.38±1.35 | 40.03±0.98 |
| OPMC | 41.25±0.40 | 47.50±0.65 | 48.25±1.95 | 62.86±1.78 | **88.51±0.57** | 41.96±1.40 | 72.31±0.24 | 26.91±0.44 | 88.00±1.10 | 40.97±1.69 |
| MVCAN | 48.32±1.28 | 45.44±2.81 | 45.56±1.67 | 17.85±0.35 | 75.63±0.14 | 21.93±0.49 | 85.57±1.14 | 19.29±0.80 | 42.95±1.14 | 38.51±1.85 |
| AE | 38.92±2.5 | 43.35±4.0 | 40.39±2.3 | 36.66±2.69 | 54.48±3.2 | 32.16±1.6 | 33.83±3.6 | 15.58±0.6 | 56.76±3.8 | 33.59±2.3 |
| DCCA-AE | 45.39±2.8 | 43.18±2.8 | 48.18±3.4 | 46.53±2.7 | 58.37±3.9 | 32.59±1.3 | 92.19±2.4 | 18.28±0.9 | 59.62±1.1 | 33.70±2.0 |
| $\ell_0$-DCCA | 41.64±2.8 | 34.44±3.69 | 39.24±2.04 | 44.46±1.18 | 47.46±2.38 | 28.10±1.74 | 99.54±0.09 | 25.41±0.83 | 42.57±2.60 | 38.67±1.88 |
| **COPER** | **49.13±3.2** | **53.15±3.3** | **54.83±3.9** | **64.65±2.6** | 82.81±5.1 | **49.80±0.8** | **99.88±0.0** | **28.06±1.1** | **92.57±0.8** | 40.68±1.6 |

Table 3: Clustering ACC evaluation on linear embedding schemes. We applied $K$-means to the Raw dataset, PCA or CCA-transformed, and CCA-transformed with permutations (CCA w perm). The ARI, and NMI evaluations are in Table 7

| Method | METABRIC | Reuters | Caltech101-20 | VOC | Caltech5V-7 | RBGD | MNIST-USPS | CCV | MSRVC1 | Scene15 |
|---|---|---|---|---|---|---|---|---|---|---|
| | | | | | ACC | | | | | |
| Raw | 35.18±2.9 | 37.16±5.9 | 42.98±3.1 | 52.41±7.3 | 73.54±6.3 | 43.15±1.8 | 69.78±6.3 | 16.25±0.7 | 74.00±7.3 | 37.91±0.9 |
| PCA | 37.72±1.4 | 42.56±2.9 | 41.47±3.4 | 43.66±4.9 | 74.26±6.3 | 42.64±1.9 | 68.14±3.9 | 16.17±0.5 | 74.14±6.1 | 37.53±1.2 |
| CCA | 40.05±2.1 | 42.85±2.5 | 42.34±3.1 | 53.1±4.1 | 75.96±5.2 | 41.47±2.2 | 80.15±5.0 | 16.16±0.7 | 73.29±7.3 | 37.59±1.3 |
| CCA w perm | **40.6±1.6** | **43.16±1.7** | **43.08±3.8** | **53.52±4.4** | **77.44±5.6** | **44.29±2.1** | **84.99±6.3** | **16.39±0.6** | **75.00±4.1** | **38.25±1.2** |

of the datasets are presented in Table B, and a complete description appears in Appendix B. The implementation details are detailed in Appendix E.1

We assess the clustering performance with three commonly used metrics: Clustering Accuracy (ACC), adjusted Rand index (ARI), and Normalized Mutual Information (NMI). ACC and ARI are scored between 0 and 1, with higher values indicating better clustering performance. We conducted each experiment 10 times and reported the mean and standard deviation for each metric. We compare our model to deep end-to-end models DSMVC (Tang & Liu, 2022), CVCL (Chen et al., 2023), ICMVC (Chao et al., 2024) and RMCNC (Sun et al., 2024), OPMC Liu et al. (2021), MVCAN Xu et al. (2024) [1] in addition to two-stage baselines, where $K$-means was applied: Autoencoder (AE) transformations; Deep CCA with autoencoders (DCCA-AE) (Chandar et al., 2016; Wang et al., 2015) and L0-CCA Lindenbaum et al. (2021) transformations. The last was successfully applied for clustering in (Benton et al., 2017; Gao et al., 2020). We present our ACC results in Table 2 and ACC, ARI, NMI in 6. Our model surpasses state-of-the-art models in both accuracy and the adjusted Rand index across all datasets. The accuracy improvement reaches up to 7% [2].

## 5.2 PERMUTATIONS ENHANCE PERFORMANCE IN LINEAR BASELINES

We now evaluate the effect of applying permutations on representations learned with linear models. We present in Table 3 ACC evaluation of $K$-means applied on raw samples (Raw) without any feature transformations, PCA transformations (PCA), linear CCA transformations (CCA), and CCA-transformed samples with permutations (CCA w perm). These results demonstrate that the clustering accuracy can be improved by introducing our within-cluster permutation procedure. We further show ARI and NMI results in Table 7.

## 5.3 ABLATION STUDY

We performed an ablation study to assess the impact of different components of our model. We created five variations of our model, each including different subsets of the components of COPER. These variations were: (i) COPER with linear encoder, (ii) COPER without the correlation loss, (iii)

---

[1]The code implementation provided by (Sun et al., 2024; Lindenbaum et al., 2021) is restricted to two views, hence we randomly chose two views from each dataset with more than two views

[2]We note that the results of Chen et al. (2023); Tang & Liu (2022) are different from the values reported by the authors. We report the mean over ten runs, while they report the best result out of ten runs. We argue that our evaluation, which includes the standard deviation, provides a more informative indication of the capabilities of each method. The best results can be found in the Appendix, Table H

COPER without the within-cluster permutations, (iv) COPER without the multi-view agreement term, and (v) COPER. We conducted this ablation study using the MSRVC1 dataset.

Table 4: Ablation study on the MSRVC1 datasets.

| Model | ACC | ARI | NMI |
|---|---|---|---|
| COPER with linear $\mathcal{F}$ | 48.00±4.6 | 22.22±5.0 | 32.26±5.3 |
| COPER w/o $\mathcal{L}_{corr}$ | 79.71±3.3 | 64.46±4.4 | 70.76±4.1 |
| COPER w/o permutations | 85.33±7.5 | 74.80±10.2 | 79.87±7.2 |
| COPER w/o multi-view agree. | 86.29±7.7 | 76.10±10.6 | 80.81±7.6 |
| COPER | 92.57±0.8 | 84.57±1.4 | 86.91±1.3 |

Table 4 presents the clustering metrics across variations of COPER. The results indicate that the correlation loss, permutations, and multi-view agreements are essential components of our model and boost the accuracy by 6-12%.

## 5.4 SCALABILITY STUDY

In this section, we assess the scalability of our model using a large-scale dataset. We generated two different views from the first 300,000 samples of the infinite MNIST dataset (Loosli et al., 2007) - one view with random background noise and the other with random background images from CIFAR10 (Krizhevsky, 2009). Table 5 shows the clustering performance of 10 experiments, along with the mean and standard deviation of each metric. Since deep learning methods are known to be suitable for large-scale datasets, we compared our model to two recent end-to-end deep MVC models: DSMVC (Chao et al., 2024) and RMCNC (Sun et al., 2024). Other methods were not included due to practical limitations imposed by the scale of the dataset [3]. Our evaluation demonstrates that our method provides significantly more accurate cluster assignments for this large and noisy dataset.

Table 5: Clustering performance metrics on large datasets generated using infinite MNIST.

| Model | ACC | ARI | NMI |
|---|---|---|---|
| DSMVC | 11.69±0.02 | 00.22±0.05 | 00.42±0.10 |
| RMCNC | 54.47±6.23 | 47.51±5.26 | 57.73±3.33 |
| COPER | 81.93±5.44 | 66.75±5.29 | 70.08±4.08 |

Results show that COPER outperforms both DSMVC (Chao et al., 2024) and RMCNC (Sun et al., 2024). Indicating that it is scalable compared to other baselines.

## 5.5 CONCLUSION AND LIMITATIONS

Our work presents a new approach called COrrelation-based PERmutations (COPER), a deep learning model for multi-view clustering (MVC). COPER integrates clustering and representation tasks into an end-to-end framework, eliminating the need for a separate clustering step. The model employs a unique self-supervision task, where within-cluster pseudo-labels are permuted across views for canonical correlation analysis loss, contributing to the maximization of between-class variance and minimization of within-class variation in the shared embedding space. We demonstrate that, under mild assumptions, our model approximates the projection achieved by linear discriminant analysis (LDA). Finally, we perform an extensive experimental evaluation showing that our model can cluster diverse data types accurately.

Our model has limitations, such as the potential need for relatively large batch sizes due to the DCCA loss (Andrew et al., 2013). This can be addressed by alternative objectives like soft decorrelation (Chang et al., 2018). Additionally, the loss combination presented here is suboptimal, and exploring smarter multitask schemes such as (Achituve et al., 2024) is a promising direction for improving the method. Other directions for future work include introducing interpretability modules (Svirsky & Lindenbaum, 2024) and extending the model to cluster data under domain shift (Rozner et al., 2023).

## 6 ACKNOWLEDGMENTS AND DISCLOSURE OF FUNDING

The work of OL is supported by the MOST grant number 0007341.

---

[3] The code provided by CVCL (Chen et al., 2023) and ICMVC (Chao et al., 2024) could not scale to this data.

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

# A  RESULTS

Table 6: Clustering evaluation using ten datasets. Our model (COPER) is compared against four recent end-to-end MVC models (DSMVC (Tang & Liu, 2022), CVCL (Chen et al., 2023), ICMVC (Chao et al., 2024), RMCNC (Sun et al., 2024), OPMC (Liu et al., 2021), MVCAN (Xu et al., 2024)), and two two-stage schemes.

| Method | METABRIC | Reuters | Caltech101-20 | VOC | Caltech5V-7 | RBGD | MNIST-USPS | CCV | MSRVC1 | Scene15 |
|---|---|---|---|---|---|---|---|---|---|---|
| | | | | | ACC | | | | | |
| DSMVC | 40.60±3.8 | 46.37±4.4 | 39.33±2.4 | 57.82±5.0 | 79.24±9.5 | 39.77±3.6 | 70.06±10.3 | 17.90±1.2 | 60.71±15.2 | 34.30±2.9 |
| CVCL | 42.66±6.2 | 45.06±8.0 | 33.50±1.4 | 36.88±3.1 | 78.58±5.0 | 31.04±1.8 | 99.38±0.1 | 26.23±1.9 | 77.90±12.3 | 40.16±1.8 |
| ICMVC | 32.12±1.16 | 38.18±0.78 | 26.54±0.52 | 36.94±1.29 | 60.97±2.37 | 32.97±1.23 | 99.29±0.08 | 20.88±1.21 | 66.28±6.11 | **41.26±0.89** |
| RMCNC | 32.55±1.30 | 37.61±1.50 | 36.73±1.38 | 39.66±1.48 | 56.77±2.98 | 33.22±1.85 | 76.37±4.13 | 21.46±0.69 | 32.38±1.35 | 40.03±0.98 |
| OPMC | 41.25±0.40 | 47.50±0.65 | 48.25±1.95 | 62.86±1.78 | **88.51±0.57** | 41.96±1.40 | 72.31±0.24 | 26.91±0.44 | 88.00±1.10 | 40.97±1.69 |
| MVCAN | 48.32±1.28 | 45.44±2.81 | 45.56±1.67 | 17.85±0.35 | 75.63±0.14 | 21.93±0.49 | 85.57±1.14 | 19.29±0.80 | 42.95±1.14 | 38.51±1.85 |
| AE | 38.92±2.5 | 43.35±4.0 | 40.39±2.3 | 36.66±2.69 | 54.48±3.2 | 32.16±1.6 | 33.83±3.6 | 15.58±0.6 | 56.76±3.8 | 33.59±2.3 |
| DCCA-AE | 45.39±2.8 | 43.18±2.8 | 48.18±3.4 | 46.53±2.7 | 58.37±3.9 | 32.59±1.3 | 92.19±2.4 | 18.28±0.9 | 59.62±1.1 | 33.70±2.0 |
| $\ell_0$-DCCA | 41.64±2.8 | 34.44±3.69 | 39.24±2.04 | 44.46±1.18 | 47.46±2.38 | 28.10±1.74 | 99.54±0.09 | 25.41±0.83 | 42.57±2.60 | 38.67±1.88 |
| **COPER** | **49.13±3.2** | **53.15±3.3** | **54.83±3.9** | **64.65±2.6** | 82.81±5.1 | **49.80±0.8** | **99.88±0.0** | **28.06±1.1** | **92.57±0.8** | 40.68±1.6 |
| | | | | | ARI | | | | | |
| DSMVC | 18.24±2.4 | **23.41±5.0** | 31.21±2.4 | 50.78±4.6 | 69.06±9.4 | 23.74±2.6 | 56.87±13.4 | 5.97±0.5 | 42.63±19.0 | 18.85±2.5 |
| CVCL | 22.69±4.3 | 22.39±8.8 | 24.85±0.9 | 22.49±2.7 | 63.25±6.3 | 16.16±1.3 | 98.63±0.2 | 12.72±1.3 | 64.27±13.8 | 24.01±1.7 |
| ICMVC | 10.46±1.21 | 16.03±0.52 | 18.00±0.50 | 25.47±0.79 | 41.57±2.51 | 15.63±0.92 | 98.42±0.18 | 5.94±0.32 | 45.73±7.23 | **25.68±0.72** |
| RMCNC | 8.94±2.39 | 10.25±0.95 | 27.18±1.47 | 18.93±1.15 | 37.33±2.47 | 15.47±0.71 | 61.66±3.24 | 8.16±0.50 | 8.79±0.97 | 24.50±0.72 |
| OPMC | 22.69±0.38 | 22.36±0.44 | 39.78±2.76 | 51.81±2.82 | **77.88±0.94** | 22.91±0.89 | 66.30±0.28 | 10.70±0.30 | 75.28±1.45 | 23.89±0.72 |
| MVCAN | **27.60±0.59** | 23.25±2.18 | 37.94±1.36 | 6.43±0.30 | 61.45±0.66 | 8.69±0.31 | 79.19±1.93 | 7.68±0.53 | 17.57±0.75 | 25.51±1.07 |
| AE | 19.71±3.0 | 10.36±4.3 | 31.27±4.1 | 18.86±2.7 | 30.73±4.8 | 15.26±1.6 | 13.93±2.0 | 3.56±0.3 | 33.43±4.8 | 17.00±1.7 |
| DCCA-AE | 21.70±2.8 | 7.63±4.0 | 37.19±3.2 | 32.77±3.4 | 34.63±4.1 | 15.00±2.0 | 84.04±4.4 | 5.0±0.4 | 38.41±2.5 | 17.12±1.6 |
| $\ell_0$-DCCA | 21.74±1.94 | 0.65±2.09 | 27.37±1.93 | 30.18±1.92 | 29.65±0.45 | 8.26±0.70 | 98.99±0.21 | 11.46±0.96 | 17.56±2.64 | 22.57±0.56 |
| **COPER** | 26.77±2.4 | 22.80±4.3 | **49.55±5.3** | **53.26±4.0** | 69.53±6.7 | **34.17±1.4** | **99.73±0.0** | **12.27±0.4** | **84.57±1.4** | 25.00±1.1 |
| | | | | | NMI | | | | | |
| DSMVC | 25.51±1.9 | 29.48±4.8 | 60.72±1.4 | **65.13±3.2** | **75.08±6.8** | 37.60±2.5 | 67.80±11.5 | 16.70±0.9 | 57.77±13.0 | 36.55±3.3 |
| CVCL | 32.3±4.5 | 29.41±11.1 | 56.13±1.0 | 32.95±1.3 | 69.56±4.8 | 26.60±1.6 | 98.21±0.3 | 26.25±0.9 | 72.66±8.9 | 41.13±1.7 |
| ICMVC | 18.90±1.02 | 20.52±0.26 | 42.90±0.59 | 42.72±0.53 | 50.82±2.10 | 26.87±0.87 | 97.93±0.22 | 15.13±0.56 | 56.44±4.37 | **43.97±0.47** |
| RMCNC | 14.00±2.00 | 17.15±1.66 | 39.21±0.64 | 23.17±0.63 | 46.24±1.58 | 28.61±0.92 | 64.49±2.30 | 16.65±0.55 | 18.81±0.96 | 41.22±0.48 |
| OPMC | 29.12±0.28 | **33.49±1.14** | **66.16±1.14** | 62.06±0.58 | **80.39±0.73** | 37.00±0.43 | 77.38±0.22 | 22.94±0.40 | 79.08±0.88 | 41.50±0.57 |
| MVCAN | **34.49±1.01** | 30.25±1.73 | 62.52±0.31 | 11.96±0.24 | 70.46±0.62 | 17.81±0.46 | 86.58±1.42 | 19.46±1.15 | 32.74±1.04 | 42.46±1.16 |
| AE | 27.34±2.6 | 18.61±3.2 | 50.15±2.4 | 47.29±4.0 | 38.97±5.2 | 28.85±1.4 | 22.83±2.5 | 10.68±1.0 | 48.31±4.0 | 33.18±2.6 |
| DCCA-AE | 30.95±2.8 | 23.04±3.1 | 54.46±2.4 | 47.29±4.0 | 43.38±3.9 | 27.54±3.5 | 85.32±3.3 | 16.39±1.0 | 52.43±4.1 | 33.65±2.0 |
| $\ell_0$-DCCA | 33.91±1.38 | 18.51±4.30 | 45.81±1.05 | 43.75±1.58 | 46.44±1.01 | 14.91±0.67 | 98.73±0.26 | **29.57±1.20** | 34.73±2.48 | 42.04±0.54 |
| **COPER** | 34.07±2.6 | 31.10±4.1 | 49.25±6.2 | 58.54±2.2 | 74.03±4.6 | **38.13±0.9** | **99.64±0.1** | 26.32±0.7 | **86.91±1.3** | 41.98±1.2 |

Table 7: Clustering evaluation on linear embedding schemes. We applied $K$-means to the Raw dataset, PCA or CCA-transformed, and CCA-transformed with permutations (CCA w perm).

| Method | METABRIC | Reuters | Caltech101-20 | VOC | Caltech5V-7 | RBGD | MNIST-USPS | CCV | MSRVC1 | Scene15 |
|---|---|---|---|---|---|---|---|---|---|---|
| | | | | | ACC | | | | | |
| Raw | 35.18±2.9 | 37.16±5.9 | 42.98±3.1 | 52.41±7.3 | 73.54±6.3 | 43.15±1.8 | 69.78±6.3 | 16.25±0.7 | 74.00±7.3 | 37.91±0.9 |
| PCA | 37.72±1.4 | 42.56±2.9 | 41.47±3.4 | 43.66±4.9 | 74.26±6.3 | 42.64±1.9 | 68.14±3.9 | 16.17±0.5 | 74.14±6.1 | 37.53±1.2 |
| CCA | 40.05±2.1 | 42.85±2.5 | 42.34±3.1 | 53.1±4.1 | 75.96±5.2 | 41.47±2.2 | 80.15±5.0 | 16.16±0.7 | 73.29±7.3 | 37.59±1.3 |
| CCA w perm | **40.6±1.6** | **43.16±1.7** | **43.08±3.8** | **53.52±4.4** | **77.44±5.6** | **44.29±2.1** | **84.99±6.3** | **16.39±0.6** | **75.00±4.1** | **38.25±1.2** |
| | | | | | ARI | | | | | |
| Raw | 18.64±1.2 | 7.49±7.7 | 34.58±4 | 29.51±7.4 | 57.19±5.9 | 24.13±1.2 | 58.88±4.8 | **5.45±0.3** | **58.53±5.8** | 22.31±0.4 |
| PCA | 19.05±1.5 | **19.88±1.4** | 31.75±3.5 | 21.44±5.1 | 57.91±4.7 | 23.78±1.4 | 54.55±2.8 | **5.45±0.2** | 58.39±5.9 | 22.0±0.4 |
| CCA | 19.84±1.7 | 18.76±1.7 | 32.68±3.8 | **34.69±5.4** | 60.01±1.9 | 24.59±1.7 | 71.09±4.5 | 5.32±0.3 | 56.97±7.2 | 22.57±0.5 |
| CCA w perm | **20.46±1.5** | 17.58±1.7 | **32.89±4.3** | 34.44±6.3 | **61.65±4.3** | **25.04±1.2** | **76.02±5.9** | 5.39±0.3 | 57.91±3.7 | **22.63±0.4** |
| | | | | | NMI | | | | | |
| Raw | 24.13±1.4 | 15.38±9.0 | **61.77±2.0** | 53.89±5.5 | 64.34±3.9 | 38.83±0.8 | 70.43±2.4 | 15.07±0.5 | **66.89±3.8** | 40.74±0.4 |
| PCA | 25.44±1.7 | **27.92±1.2** | 60.50±1.4 | 44.29±3.9 | 63.79±2.5 | 37.43±1.0 | 65.27±1.1 | **15.17±0.3** | 67.37±4.0 | 40.33±0.5 |
| CCA | 26.85±2.0 | 27.16±1.0 | 61.13±2.0 | 53.01±3.6 | 65.72±1.9 | 38.79±1.0 | 77.58±2.2 | 14.38±0.4 | 65.52±5.2 | 40.75±0.5 |
| CCA w perm | **27.82±1.6** | 26.96±0.6 | 61.08±2.2 | **54.54±3.3** | **66.89±2** | **39.32±0.8** | 79.94±2.7 | 14.16±0.4 | 65.89±3.1 | **41.02±0.5** |

# B  DATASETS DESCRIPTION

- **METABRIC** Curtis et al. (2012): Consists of $1,440$ samples from breast cancer patients, which are annotated by 8 subtypes based on InClust Dawson et al. (2013). We observe two modalities, namely the RNA gene expression data and Copy Number Alteration (CNA) data. The dimensions of these modalities are $15,709$ and $47,127$, respectively.

- **Reuters** Amini et al. (2009): Consists of $18,758$ documents from 6 different classes. Documents are represented as a bag of words using a TFIDF-based weighting scheme. This dataset is a subset of the Reuters database, comprising the English version as well as translations in four distinct languages: French, German, Spanish, and Italian. Each language is treated as a different view. To further reduce the input dimensions, we preprocess the data with a truncated version of $SVD$, turning all input dimensions to $3,000$.

- **Caltech101-20** Zhao et al. (2017): Consists of $2,386$ images of 20 classes. This dataset is a subset of Caltech101. Each view is an extracted handcrafted feature, including the

Gabor feature, Wavelet Moments, CENTRIST feature, HOG feature, GIST feature, and LBP feature. [4].

- **VOC** Everingham et al. (2010): Consists of $9,963$ image and text pairs from 20 different classes. Following the conventions by Trosten et al. (2021a); Van der Maaten & Hinton (2008), $5,649$ instances are selected to construct a two-view dataset, where the first and the second view is 512 Gist features and 399 word frequency count of the instance, respectively.

- **Caltech-5V-7** Dueck & Frey (2007): Consists of $1,400$ images of 7 classes. Same as Caltech101-20, this dataset is also a subset of Caltech101 and is comprised of the same views apart from the Gabor feature. [4].

- **RBGD** Kong et al. (2014): Consists of $1,449$ samples of indoor scenes image-text of 13 classes. We follow the version provided in Trosten et al. (2021a); Zhou & Shen (2020), where image features are extracted from a ResNet50 model pre-trained on the ImageNet dataset and text features from a doc2vec model pre-trained on the Wikipedia dataset.

- **MNIST-USPS** Asuncion & Newman (2007): Consists of $5,000$ digits from 10 different classes (digits). MNIST and USPS are both handwritten digital datasets and are treated as two different views.

- **CCV** Jiang et al. (2011b): Consists of $6,773$ samples of indoor scenes image-text of 20 classes. Following the convention in Li et al. (2019b) we use the subset of the original CCV data. The views comprise of three hand-crafted features: STIP features with $5,000$ dimensional Bag-of-Words (BoWs) representation, SIFT features extracted every two seconds with $5,000$ dimensional BoWs representation, and MFCC features with $4,000$ dimensional BoWs.

- **MSRCv1** Consists of 210 scene recognition images belonging to 7 categories Zhao et al. (2020). Each image is described by five different types of features.

- **Scene15** Consists of 4,485 scene images belonging to 15 classes Fei-Fei & Perona (2005).

- **300K-MNIST-CIFAR10** Loosli et al. (2007); Krizhevsky (2009): Consists of $300,000$ samples generated from the infinite MNIST dataset. Two different views are created: one with random background noise and the other with random background images from CIFAR10. This dataset is used to assess the scalability and robustness of clustering methods under noisy and large-scale conditions. Deep learning-based models such as DSMVC Chao et al. (2024) and RMCNC Sun et al. (2024) demonstrate improved performance over traditional approaches on this dataset.

Table 8: Datasets used in our experiments.

| Dataset | # Samples | # Classes (K) | # Views | Dimensions | Ref |
|---|---|---|---|---|---|
| METABRIC | 1440 | 8 | 2 | [15709, 47127] | Curtis et al. (2012) |
| Reuters | 18758 | 6 | 5 | [21531, 24892, 34251, 15506, 11547] | Amini et al. (2009) |
| Caltech101-20 | 2386 | 20 | 6 | [48, 40, 254, 1984, 512, 928] | Zhao et al. (2017) |
| VOC | 5649 | 20 | 2 | [512, 399] | Everingham et al. (2010) |
| Caltech-5V-7 | 1400 | 7 | 5 | [40, 254, 1984, 512, 928] | Dueck & Frey (2007) |
| RBGD | 1449 | 13 | 2 | [2048, 300] | Kong et al. (2014) |
| MNIST-USPS | 5000 | 10 | 2 | [784, 784] | Asuncion & Newman (2007) |
| CCV | 6773 | 20 | 3 | [4000, 5000, 5000] | Jiang et al. (2011b) |
| MSRVC1 | 210 | 7 | 5 | [24, 576, 512, 256, 254] | Zhao et al. (2020) |
| Scene15 | 4485 | 15 | 3 | [20, 59, 40] | Fei-Fei & Perona (2005) |

## C  COMPLEXITY ANALYSIS

Let $N_{mb}$ represent the number of samples in a mini batch and $N$ the total number of samples in the dataset, $L$ denote the maximum number of neurons in the model's hidden layers across all views, and $l$ indicates the dimension of the low-dimensional embedding space. Additionally, $L'$ stands for the number of neurons in the cluster head, and the number of clusters is denoted as $K$. For reliable labels, we denote $\widetilde{N}_{mb}$ as the number of samples in the permuted mini batch.

The time complexity of training each autoencoder is $\mathcal{O}(L \cdot N_{mb} \cdot \lceil \frac{N}{N_{mb}} \rceil)$. With $M$ views, the total complexity for the pretraining phase is $\mathcal{O}(M \cdot L \cdot N \cdot \lceil \frac{N}{N_{mb}} \rceil)$. Calculating the CCA loss for each

---

[4] The creation of both Caltech101-20 and Caltech-5V-7 is due to the unbalanced classes in Caltech-101.

pair of views has a complexity of $\mathcal{O}(N_{mb}^2 \cdot \lceil \frac{N}{N_{mb}} \rceil)$ for each pair. Since there are $\binom{M}{2}$ possible pairs, the total complexity for the CCA loss is $\mathcal{O}(M^2 \cdot N_{mb}^2 \cdot \lceil \frac{N}{N_{mb}} \rceil)$.

The time complexity for training the cluster head is $\mathcal{O}(L' \cdot N_{mb} \cdot \lceil \frac{N}{N_{mb}} \rceil)$. Computing reliable labels for each view and each cluster also involves a complexity of $\mathcal{O}(M \cdot N_{mb} \cdot \lceil \frac{N}{N_{mb}} \rceil \cdot K)$. Therefore, the total complexity for the multi-view reliable labels tuning phase is $\mathcal{O}(M \cdot L' \cdot N_{mb} \cdot \lceil \frac{N}{N_{mb}} \rceil \cdot K)$.

Computing CCA loss for reliable labels involves a complexity of $\mathcal{O}(\widetilde{N}_{mb}^2 \cdot \lceil \frac{N}{\widetilde{N}_{mb}} \rceil)$. The overall time complexity of our multi-view CCC model is the sum of the complexities of the individual phases. Therefore, the total time complexity is:

$$\mathcal{O}(M \cdot L \cdot N_{mb} \cdot \lceil \frac{N}{N_{mb}} \rceil + M^2 \cdot N_{mb}^2 \cdot \lceil \frac{N}{N_{mb}} \rceil + M \cdot L' \cdot N_{mb} \cdot \lceil \frac{N}{N_{mb}} \rceil \cdot K + \widetilde{N}_{mb}^2 \cdot \lceil \frac{N}{\widetilde{N}_{mb}} \rceil).$$

Where the dominant factor is:

$$\mathcal{O}(M^2 \cdot N_{mb}^2 \cdot \lceil \frac{N}{N_{mb}} \rceil).$$

## D  MULTI-VIEW PSEUDO-LABELING

### D.1  DETAILED STEPS

**Labels prediction**  Our cluster head $\mathcal{G}$ accepts a fusion of latent embeddings. The fusion is a weighted sum $\sum_v w_v \boldsymbol{H}^{(v)}$ where $\{w_v\}_1^{n_v}$ are learnable weights. For all samples, $\mathcal{G}$ predicts clusters probabilities $\mathcal{G}(\sum_v w_v \boldsymbol{H}^{(v)}) = \boldsymbol{P} \in \mathbb{R}^{N \times K}$ where each row is a probability vector $\boldsymbol{p}_i$ for samples $\boldsymbol{x}_i^{(v)}$, for all $v \in 1, .., n_v$ (as $\boldsymbol{P}$ is shared between all views). Each cluster is represented by column index $k$ in $\boldsymbol{P}$, and by selecting the top $B$ probabilities, which are the most confident samples for each $k$ we form $\mathcal{T}_k = \{i | i \in \text{argtopk}(\boldsymbol{P}_{:,k}, B), \forall i = 1, 2, ..., N\}$. We denote $\mathcal{T} = \cup_k \mathcal{T}_k$ as the set of samples that were assigned with at least one label. At this stage, we note that (a) samples that are assigned to the same cluster based on $\mathcal{T}$ can have embedding vectors that have different geometric relations to cluster centers in different views, and (b) some samples may be assigned to more than one cluster. Therefore, we present the following steps that refine the list of pseudo-labels identified in $\mathcal{T}$.

**Labels refinement**  To deal with (a), we refine the clusters separately for each view . We start by computing the view specific cluster centers in the embedded space $\bar{\boldsymbol{h}}_k^{(v)} = 1/B \sum_{i \in \mathcal{T}_k} \boldsymbol{h}_i^{(v)}$, and the semantic similarity between them and all other embedded samples $\boldsymbol{h}_i^{(v)}, \forall i \in \mathcal{T}_k$. Semantic similarity is expressed by cosine similarity $s_{i,k} = \boldsymbol{h}_i^{(v)} \cdot \bar{\boldsymbol{h}}_k^{(v)} / \|\boldsymbol{h}_i^{(v)}\| \|\bar{\boldsymbol{h}}_k^{(v)}\|$. For each cluster, we preserve samples that share a similarity greater than a predefined threshold $\lambda$. Next, for each remaining sample, we construct a corresponding pseudo-label vector $\hat{\boldsymbol{y}}_i^{(v)}$. If a sample is assigned to a single pseudo-label then $\hat{\boldsymbol{y}}_i^{(v)}$ is a one hot vector, otherwise $\hat{\boldsymbol{y}}_i^{(v)}$ is a multi pseudo-label vector where values for the assigned pseudo-labels are $s_{i,k}$ if $s_{i,k} \geq \lambda$ and 0 otherwise. Multi pseudo-label vectors are then normalized to obtain probability vectors. Due to label filtering and issue (b), different viewpoints may now disagree on cluster assignments.

**Multi-view agreements**  We now establish an agreement between views for cluster assignments. For each $i \in \mathcal{T}$, if a sample is assigned to at least one pseudo-label in more than one view, it will be kept only if the views form agreement expressed by $\text{argmax}(\boldsymbol{p}_i^{(v)}) = \text{argmax}(\boldsymbol{p}_i^{(w)})$, and otherwise discarded. If a sample is assigned in one view only, it will be retained for training since the cluster head optimization is performed for each view separately, as described below.

**View specific probabilities**  Since our optimization is performed on each view, we compute view-specific probabilities matrices $\boldsymbol{P}^{(v)}$ by feeding $\boldsymbol{H}^{(v)}$ to $\mathcal{G}$.

The probabilities for remaining samples in $\mathcal{T}$, $\boldsymbol{p}_i^{(v)}$ and their corresponding pseudo-labels $\hat{\boldsymbol{y}}_i^{(v)}$, are used to train the model. We denote $\mathcal{X}^{(v)} = \{(\boldsymbol{x}_i, \hat{\boldsymbol{y}}_i^{(v)})\}_{i \in \mathcal{T}}$ as the set of samples with corresponding pseudo-labels in view $v$.

## D.2 EXAMPLE

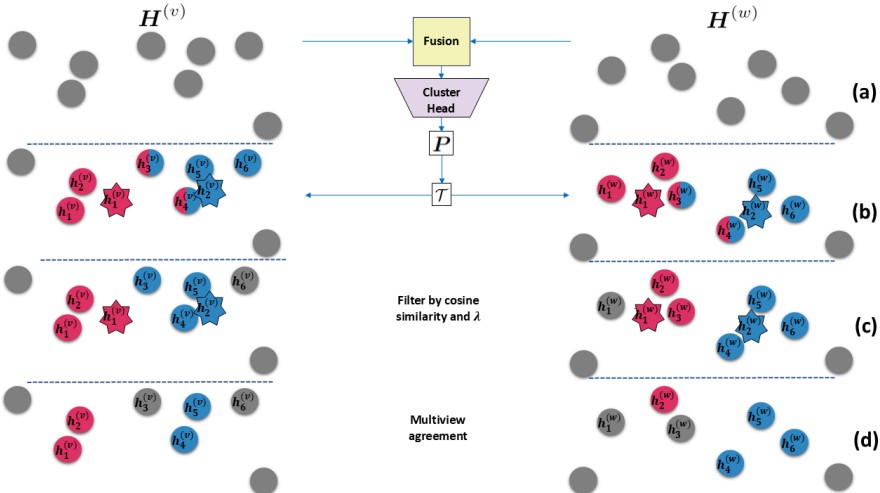

Figure 5: Illustration of our pseudo-labeling scheme.

Figure 5 illustrates our pseudo-labeling scheme. As shown in part (a), the embeddings $\boldsymbol{H}^{(v)}$ and $\boldsymbol{H}^{(w)}$ are first fused and fed into the clustering head $\mathcal{G}$ to predict clusters probabilities $\mathcal{G}(\sum_v w_v \boldsymbol{H}^{(v)}) = \boldsymbol{P}$. Next, (b), the top $B$ (in this case, 4) probabilities are selected for each of the two clusters: $\mathcal{T}_k = \{i | i \in \text{argtopk}(\boldsymbol{P}_{:,k}, B)$, where $\mathcal{T} = \cup_k \mathcal{T}_k$, and the centers of the clusters are calculated by: $\bar{\boldsymbol{h}}_k^{(v)} = 1/B \sum_{i \in \mathcal{T}_k} \boldsymbol{h}_i^{(v)}$. Note that the samples $\boldsymbol{h}_3^v, \boldsymbol{h}_3^w$ and $\boldsymbol{h}_4^v, \boldsymbol{h}_4^w$ have been assigned to more than one pseudo-label. In the next step, see part (c), we filter samples if their cosine similarity from the cluster center $s_{i,k} = \boldsymbol{h}_i^{(v)} \cdot \bar{\boldsymbol{h}}_k^{(v)} / \|\boldsymbol{h}_i^{(v)}\| \|\bar{\boldsymbol{h}}_k^{(v)}\|$ does not surpass a predefined threshold $\lambda$. Here for view $v$ (left) sample $\boldsymbol{h}_6^v$ was filtered from the blue cluster and samples $\boldsymbol{h}_3^v, \boldsymbol{h}_4^v$ were filtered only from the red cluster. For view $w$ (right) sample $\boldsymbol{h}_1^v$ was filtered from the red cluster and sample $\boldsymbol{h}_3^v$ was filtered, but only from the blue cluster. Next, we achieve multi-view agreement by filtering out $\boldsymbol{h}_3^v$ and $\boldsymbol{h}_3^w$ since they have different labels. Despite that $\boldsymbol{h}_6^v, \boldsymbol{h}_6^w$ and $\boldsymbol{h}_1^v, \boldsymbol{h}_1^w$ do not have an agreement, $\boldsymbol{h}_6^w$ and $\boldsymbol{h}_1^v$ are retained since the cross-entropy optimization is performed on each view separately. They will later be retained for within-cluster permutation only if the pseudo-label agrees with $\text{argmax}(\boldsymbol{p}_6)$ and $\text{argmax}(\boldsymbol{p}_1)$.

## E EXPERIMENTS

### E.1 IMPLEMENTATION DETAILS

We implement our model using PyTorch, and the code is available for public use [5]. All experiments were conducted using an Nvidia A100 GPU server with Intel(R) Xeon(R) Gold 6338 CPU @ 2.00GHz. The training is done with Adam optimizer with learning rate $10^{-4}$ and its additional default parameters in Pytorch.

To improve convergence stability, we add decoder modules defined for each view that reconstruct the original samples and are optimized jointly with the main model by adding mean squared error objective in addition to $\mathcal{L}_{\text{corr}}$.

### E.2 HYPERPARAMETERS TUNING

To tune the parameters for each dataset, we utilize the Silhouette Coefficient Rousseeuw (1987) as an unsupervised metric for cluster separability. The Silhouette Coefficient is calculated using each sample's mean within-cluster distance and the mean nearest-cluster distance. The distances are calculated on the fusion of the learned representations $\frac{1}{n_v} \sum_v \boldsymbol{H}^{(v)}$. We calculate the Silhouette Coefficient after each epoch and pick the configuration that produces the maximal average Silhouette

---

[5]The code will be released at Github.

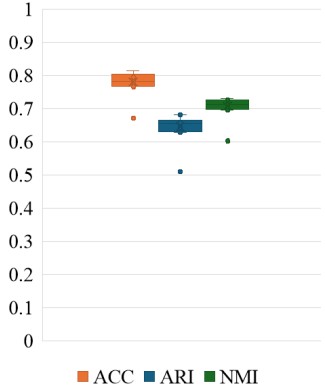

Figure 6: Box-plot for measured ACC, ARI, and NMI metrics on model trained with different batch sizes on MSRCv1 dataset.

Coefficient value across a limited set of options. We present the correlation between the Silhouette Coefficient value and clustering accuracy metrics in Table E.2 on datasets VOC and METABRIC. In addition, we set the $k$ for the argtopk function to be the batch size divided by a number of clusters. In practice, we observed that for some datasets, the parameter value should be increased due to the filtration we apply on pseudo labels. We use a fixed cosine similarity threshold for all datasets $0.5$.

Table 9: Silhouette Coefficient for different batch sizes.

| Batch Size | ACC | ARI | NMI | Silhouette Coefficient |
|---|---|---|---|---|
| VOC | | | | |
| 256 | 58.77 | 43.24 | 57.19 | 0.063 |
| 360 | **62.45** | **52.06** | **59.3** | **0.161** |
| 500 | 60.4 | 47.33 | 56.06 | 0.138 |
| METABRIC | | | | |
| 128 | 45.5 | 23.7 | 34.1 | 0.0396 |
| 256 | 53.9 | 27.8 | 37.7 | 0.0163 |
| 360 | 49.2 | 23.1 | 35.3 | 0.0569 |
| 500 | 46.4 | **27.9** | **38.9** | **0.1088** |

### E.3  $\lambda$ SENSITIVITY ANALYSIS

In our experiments, we initially set the default $\lambda = 0.5$. For three datasets (Scene15, Reuters, and CCV), we have decided to increase it after observing the convergence of cross-entropy loss. Furthermore, we provide an analysis of our chosen $\lambda$'s sensitivity on the Caltech5V dataset in Table E.3 below:

Table 10: Performance metrics (ACC, ARI, NMI) for different values of $\lambda$.

| $\lambda$ | ACC (STD) | ARI (STD) | NMI (STD) |
|---|---|---|---|
| 0 | $80.94 \pm 4.3$ | $67.12 \pm 3.9$ | $71.63 \pm 2.8$ |
| 0.45 | $82.19 \pm 2.5$ | $69.08 \pm 0.9$ | $74.57 \pm 2.9$ |
| 0.5 | $82.81 \pm 5.1$ | $69.53 \pm 6.7$ | $74.03 \pm 4.6$ |
| 0.55 | $82.81 \pm 5.1$ | $69.53 \pm 6.7$ | $74.03 \pm 4.6$ |
| 0.85 | $76.56 \pm 5.6$ | $60.16 \pm 6.7$ | $66.64 \pm 5.6$ |

This analysis shows that thresholding the pseudo-labels could have a positive impact up to some value, although a too high threshold may have a negative impact.

In addition, we provide a sensitivity analysis of the batch size hyperparameter. We train our model with batch sizes $[128, 256, 512, 1024]$ on the MSRCv1 dataset, 5 times for each batch size with different random initialization seeds. We present the box plot in Figure E.3.

### E.4 GRADUAL TRAINING

We train the model by gradually introducing additional loss terms during the training. We allow training the model with additional decoders and reconstruction loss that could enhance model stability between different random initialization seeds. We start with $\mathcal{L}_{corr}$ loss and optionally with reconstruction loss $\mathcal{L}_{mse}$. Next, after a few epochs, we add cross entropy loss $\mathcal{L}_{ce}$ being minimized with predicted with pseudo labels. Finally, we introduce the within-cluster permutations, and the model is optimized with all loss terms during the next epochs. In order to tune the number of epochs for each step, we start with 100 epochs for the first step, 50 epochs for the second step, and a total of 1000 epochs for training in total. During the experiments, we found that some of the datasets could be trained with fewer epochs by analyzing the training loss dynamics.

### E.5 NEURAL NETWORKS ARCHITECTURES

We present in Table E.5 the dimensions of the fully connected non-linear neural networks for each dataset. In case the decoder is applied, its architecture is a mirrored version of the encoder. The clustering head accepts the same dimension as the last encoder dimension, and we present only the single hidden dimension we use for each dataset.

Table 11: Model architecture for different datasets used in our experiments.

| Dataset | $\mathcal{F}$ Hidden Dimensions | $\mathcal{G}$ Hidden Dimensions |
|---|---|---|
| METABRIC | [512, 2048, 128] | [2048] |
| Reuters | [512, 512, 1024, 10] | [1024] |
| Caltech101-20 | [512, 512, 1024, 20] | [1024] |
| VOC | [512, 512, 2048, 20] | [1024] |
| Caltech-5V-7 | [512, 256, 128] | [1024] |
| RBGD | [512, 2048, 128] | [1024] |
| MNIST-USPS | [1024, 512, 128] | [1024] |
| CCV | [1024, 2048, 128] | [2048] |
| MSRVC1 | [256, 512, 128] | [1024] |
| Scene15 | [256, 512, 1024, 2048, 128] | [1024] |

## F CASE STUDY USING FASHION MNIST

We conduct a controlled experiment using F-MNIST (Xiao et al., 2017). First, we create two coupled views by horizontally splitting the images. CCA is subsequently performed on the multi-view dataset, with different versions of within-cluster sample permutations.

We only permute samples with the same label. In Fig. 7, we confirmed our hypothesis that these permutations decrease the mean within-class correlation across views (please refer to the second paragraph in Section 4.3 for more details).

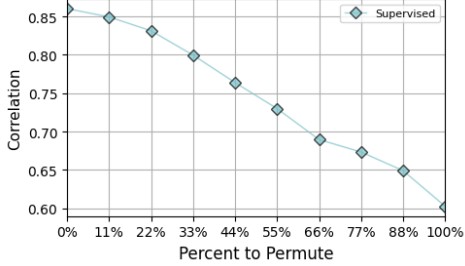

Figure 7: Permutations decrease the mean within-class correlation across views.

## G   RELATION BETWEEN COPER AND LDA

### G.1   REMINDER: LINEAR DISCRIMINANT ANALYSIS (LDA)

For a dataset $\boldsymbol{X} \in \mathbb{R}^{N \times D}$ and it's covariance matrix $\boldsymbol{C}$, we denote the within-class covariance matrix as $\boldsymbol{C_e}$ and the between-cluster covariance matrix as $\boldsymbol{C_a}$:

$$\boldsymbol{C_e} = \frac{1}{N} \sum_{k=1}^{K} \sum_{i=1}^{N_k} (\boldsymbol{x}_i^k - \boldsymbol{\mu}_k)(\boldsymbol{x}_i^k - \boldsymbol{\mu}_k)^T.$$

Where $N_k$ are the samples from class $k$, $\boldsymbol{x}_i^k$ is the $i$'th sample, and $\boldsymbol{\mu}_k$ is the mean. The between-class covariance is:

$$\boldsymbol{C_a} = \sum_{k=1}^{K} \frac{N_k}{N} \boldsymbol{\mu}_k {\boldsymbol{\mu}_k}^T,$$

and $\boldsymbol{C} = \boldsymbol{C_e} + \boldsymbol{C_a}$. The optimization for LDA can be formulated as:

$$\max_{\boldsymbol{h} \neq \boldsymbol{0}} \frac{\boldsymbol{h}^T \boldsymbol{C_a} \boldsymbol{h}}{\boldsymbol{h}^T \boldsymbol{C_e} \boldsymbol{h}}.$$

This could be solved using a generalized eigenproblem:

$$\boldsymbol{C_a} \boldsymbol{h} = \lambda \boldsymbol{C_e} \boldsymbol{h}, \quad \boldsymbol{C_e^{-1}} \boldsymbol{C_a} \boldsymbol{h} = \lambda \boldsymbol{h}.$$

### G.2   FROM COPER TO LDA

$\theta$ is treated as two different views $\theta^1$ and $\theta^2$ where each view is comprised of the same samples from $\theta$ but different, within-class permutations.

To create the views, different permutations $\Pi^l$ of $\theta$ for $l = 0, 1, 2 \ldots, \infty$ are stacked, where the order of stacked permutations is different for the two different views.

**Lemma G.1** *Applying CCA on $\theta^1$ and $\theta^2$ produces the same projection as applying LDA on $\theta$ with the (unknown) labels.*

Since both $\theta^1$ and $\theta^2$ are comprised of the same samples, it follows:

$$\boldsymbol{C_\theta} = \boldsymbol{C_{\theta^1}} = \boldsymbol{C_{\theta^2}}, \tag{4}$$

and

$$\boldsymbol{C_{\theta^1 \theta^2}} = \boldsymbol{C_{\theta_a}}, \tag{5}$$

we can use equations in 2.1 to find to solution for CCA:

$$\boldsymbol{C_\theta^{-1}} \boldsymbol{C_{\theta_a}} \boldsymbol{C_\theta^{-1}} \boldsymbol{C_{\theta_a}} \boldsymbol{h}_{CCA} = \lambda_{CCA} \boldsymbol{h}_{CCA}, \tag{6}$$

while from equation 2 we know that the solution for LDA is:

$$\boldsymbol{C_a} \boldsymbol{h}_{LDA} = \lambda_{LDA} \boldsymbol{C_e} \boldsymbol{h}_{LDA}.$$

We can further plug equation 2 and get:

$$\boldsymbol{C_a} \boldsymbol{h}_{LDA} = \lambda_{LDA} (\boldsymbol{C} - \boldsymbol{C_a}) \boldsymbol{h}_{LDA}$$

$$\lambda_{LDA} (\boldsymbol{C} - \boldsymbol{C_a}) \boldsymbol{h}_{LDA} = \lambda_{LDA} \boldsymbol{C} \boldsymbol{h}_{LDA} - \lambda_{LDA} \boldsymbol{C_a} \boldsymbol{h}_{LDA}$$

$$(1 + \lambda_{LDA}) \boldsymbol{C_a} \boldsymbol{h}_{LDA} = \lambda_{LDA} \boldsymbol{C} \boldsymbol{h}_{LDA}$$

$$\boldsymbol{C_a} \boldsymbol{h}_{LDA} = \frac{\lambda_{LDA}}{1 + \lambda_{LDA}} \boldsymbol{C} \boldsymbol{h}_{LDA}$$

$$\boldsymbol{C}^{-1} \boldsymbol{C_a} \boldsymbol{h}_{LDA} = \frac{\lambda_{LDA}}{1 + \lambda_{LDA}} \boldsymbol{h}_{LDA},$$

Next, if we substitute $\boldsymbol{C}^{-1} \boldsymbol{C_a} = A$ and $\frac{\lambda_{LDA}}{1 + \lambda_{LDA}} = \lambda^*$ in G.2 we get :

$$A \boldsymbol{h}_{LDA} = \lambda^* \boldsymbol{h}_{LDA}, \tag{7}$$

In addition if we substitute this in 6 we get:

$$\boldsymbol{C_\theta}^{-1} \boldsymbol{C_{\theta_a}} \boldsymbol{C_\theta}^{-1} \boldsymbol{C_{\theta_a}} h_{CCA} = AAh_{CCA}$$
$$AAh_{CCA} = A(Ah_{CCA})$$
$$A(Ah_{CCA}) = A(\lambda^* \boldsymbol{h}_{LDA})$$
$$A(\lambda^* \boldsymbol{h}_{LDA}) = \lambda^* (A\boldsymbol{h}_{LDA})$$
$$\lambda^* (A\boldsymbol{h}_{LDA}) = \lambda^* (\lambda^* \boldsymbol{h}_{LDA})$$
$$\lambda^* (\lambda^* \boldsymbol{h}_{LDA}) = \lambda_{CCA} \boldsymbol{h}_{CCA},$$

Since the canonical correlation coefficients are the square root of the eigenvalue obtained from the generalized eigenvalue problem, it follows that $\boldsymbol{h}_{LDA} = \boldsymbol{h}_{CCA}$.

### G.3 LDA APPROXIMATION

First, to simplify notation, we denoted $\boldsymbol{h}$ and $\hat{\boldsymbol{h}}$ the LDA representation and our representation, which is based on pseudo-labels that are potentially incorrect. In the previous sections we saw that $\hat{\boldsymbol{C}}_{\boldsymbol{\theta}}^{-1}$ and $\hat{\boldsymbol{C}}_{\boldsymbol{\theta}_a}$ are used compute $\boldsymbol{h}$ in equation 2. Hence, to assess this equivalence, we draw attention to potential errors in estimating $\hat{\boldsymbol{C}}_{\boldsymbol{\theta}}$ from equation 4, where $\hat{\boldsymbol{C}}_{\boldsymbol{\theta}} = \hat{\boldsymbol{C}}_{\boldsymbol{\theta}_e} + \hat{\boldsymbol{C}}_{\boldsymbol{\theta}_a}$.

Let $\hat{N}_k$ be the estimated samples for each class $k$, and $\hat{\boldsymbol{\mu}}_k$ it's estimated mean. $\bar{N}_k$ are the indices of samples from class $k$ not included in $\hat{N}_k$. $\tilde{N}_k$ are samples not in class $k$, which are incorrectly included in $\hat{N}_k$, and $\ddot{N}_k$ are samples in $\hat{N}_k$ which are correctly included, the corresponding class for samples in $\dot{N}_k$ are assumed to be known for this analysis.

For $\hat{\boldsymbol{C}}_{\boldsymbol{\theta}_e} = \frac{1}{N_{mb}} \sum_{k=1}^{K} (\boldsymbol{x}_i^k - \hat{\boldsymbol{\mu}}_k)(\boldsymbol{x}_i^k - \hat{\boldsymbol{\mu}}_k)^T$, Incomplete inclusion of all between-class samples may cause error. As this depends on the batch size and the pseudo-labels batch size. We denote this type of error as:

$$\boldsymbol{E}^1 = -\frac{1}{\sum_{k=1}^{K} |\bar{N}_k|} \sum_{k=1}^{K} \sum_{i \in \bar{N}_k} (\boldsymbol{x}_i^k - \hat{\boldsymbol{\mu}}_k)(\boldsymbol{x}_i^k - \hat{\boldsymbol{\mu}}_k)^T.$$

In addition, false pseudo-labels cause samples from different classes to be permuted together. We denote this type of error as:

$$\boldsymbol{E}^2 = \frac{1}{\sum_{k=1}^{K} |\tilde{N}_k||\ddot{N}_k|} \sum_{k=1}^{K} \sum_{i \in \ddot{N}_k} \sum_{j \in \tilde{N}_k} (\boldsymbol{x}_i^k - \hat{\boldsymbol{\mu}}_k)(\boldsymbol{x}_j^q - \hat{\boldsymbol{\mu}}_q)^T.$$

where $q$ is the true class of samples $j$.

For $\hat{\boldsymbol{C}}_{\boldsymbol{\theta}_a} = \frac{\hat{N}_k}{N_{mb}} \sum_{k=1}^{K} \hat{\boldsymbol{\mu}}_k (\hat{\boldsymbol{\mu}}_k)^T$. Errors in estimating $\hat{\mu}_k$, denoted by $\Delta \boldsymbol{\mu}_k = \boldsymbol{\mu}_k - \hat{\boldsymbol{\mu}}_k$ can be further propagated to the third type of error, denoted as:

$$\boldsymbol{E}^3 = -\frac{\hat{N}_k}{N_{mb}} \Delta \boldsymbol{\mu}_k (\Delta \boldsymbol{\mu}_k)^T.$$

Together all three types of errors can be expressed as $\boldsymbol{E} := \boldsymbol{E}^1 + \boldsymbol{E}^2 + \boldsymbol{E}^3$:

Taking these errors into account, we can now treat them as a perturbation. Formally:

$$\hat{\boldsymbol{C}}_{\boldsymbol{\theta}_a} = \boldsymbol{C}_{\boldsymbol{\theta}_a} + \boldsymbol{E}^3,$$
$$\hat{\boldsymbol{C}}_{\boldsymbol{\theta}_e} = \boldsymbol{C}_{\boldsymbol{\theta}_e} + \boldsymbol{E}^1 + \boldsymbol{E}^2,$$
$$\hat{\boldsymbol{C}}_{\boldsymbol{\theta}} = \boldsymbol{C}_{\boldsymbol{\theta}} + \boldsymbol{E}.$$

Since the latent dimension is significantly smaller than the input dimension, it is plausible to assume that $\hat{\boldsymbol{C}}_{\boldsymbol{\theta}}$ is invertible. In addition, we express the first order approximation of the inverse as:

$$\hat{\boldsymbol{C}}_{\boldsymbol{\theta}}^{-1} = (\boldsymbol{C}_{\boldsymbol{\theta}} + \boldsymbol{E})^{-1} = \boldsymbol{C}_{\boldsymbol{\theta}}^{-1} - \boldsymbol{C}_{\boldsymbol{\theta}}^{-1} \boldsymbol{E} \boldsymbol{C}_{\boldsymbol{\theta}}^{-1}.$$

This is accurate in terms of order $||E||^2$ Stewart & Sun (1990).

This means that the estimated matrix $A$ from Eq. 7, $\hat{A}$ can be written as:

$$\hat{A} = \hat{C}_\theta^{-1}\hat{C}_{\theta_a}$$
$$\hat{C}_\theta^{-1}\hat{C}_{\theta_a} = (C_\theta^{-1} - C_\theta^{-1}EC_\theta^{-1})(C_{\theta_a} + E^3)$$
$$(C_\theta^{-1} - C_\theta^{-1}EC_\theta^{-1})(C_{\theta_a} + E^3) = (C_\theta^{-1} - C_\theta^{-1}EC_\theta^{-1})(C_\theta - C_{\theta_e} + E^3)$$
$$(C_\theta^{-1} - C_\theta^{-1}EC_\theta^{-1})(C_\theta - C_{\theta_e} + E^3) = C_\theta^{-1}C_\theta - C_\theta^{-1}EC_\theta^{-1}C_\theta -$$
$$\quad C_\theta^{-1}C_{\theta_e} + C_\theta^{-1}EC_\theta^{-1}C_{\theta_e} + C_\theta^{-1}E^3 - C_\theta^{-1}EC_\theta^{-1}E^3$$
$$C_\theta^{-1}C_\theta - C_\theta^{-1}EC_\theta^{-1}C_\theta - C_\theta^{-1}C_{\theta_e} +$$
$$\quad C_\theta^{-1}EC_\theta^{-1}C_{\theta_e} + C_\theta^{-1}E^3 - C_\theta^{-1}EC_\theta^{-1}E^3 =$$
$$\quad I - C_\theta^{-1}EI - C_\theta^{-1}C_{\theta_e} + C_\theta^{-1}EC_\theta^{-1}C_{\theta_e} + C_\theta^{-1}E^3 - C_\theta^{-1}EC_\theta^{-1}E^3,$$

and $A$ can be expressed as:

$$A = C_\theta^{-1}C_{\theta_a},$$
$$C_\theta^{-1}C_{\theta_a} = C_\theta^{-1}(C_\theta - C_{\theta_e}),$$
$$C_\theta^{-1}(C_\theta - C_{\theta_e}) = C_\theta^{-1}C_\theta - C_\theta^{-1}C_{\theta_e},$$
$$C_\theta^{-1}C_\theta - C_\theta^{-1}C_{\theta_e} = I - C_\theta^{-1}C_{\theta_e}.$$

Now we can estimate the perturbation from $A$, denoted as $D$:

$$D = A - \hat{A},$$
$$A - \hat{A} =$$
$$= (I - C_\theta^{-1}C_{\theta_e}) - (I - C_\theta^{-1}EI - C_\theta^{-1}C_{\theta_e} + C_\theta^{-1}EC_\theta^{-1}C_{\theta_e} + C_\theta^{-1}E^3 - C_\theta^{-1}EC_\theta^{-1}E^3),$$
$$(I - C_\theta^{-1}C_{\theta_e}) - (I - C_\theta^{-1}EI - C_\theta^{-1}C_{\theta_e} + C_\theta^{-1}EC_\theta^{-1}C_{\theta_e} + C_\theta^{-1}E^3 - C_\theta^{-1}EC_\theta^{-1}E^3) =$$
$$\quad C_\theta^{-1}E - C_\theta^{-1}EC_\theta^{-1}C_{\theta_e} - C_\theta^{-1}E^3 + C_\theta^{-1}EC_\theta^{-1}E^3.$$

This perturbation can be used as a bound for the approximated eigenvalues according to Stewart & Sun (1990):

$$|\hat{\lambda}_i - \lambda_i| \leq ||D||_2, i = 1 \ldots n. \tag{8}$$

## H COMPARING THE BEST RESULTS

In the Table H, we provide the results obtained by our model in the same setup as in previous works where the best accuracy values are selected from multiple runs with random initialization.

Table 12: Best clustering results. Our model (COPER) is compared against two recent deep MVC models.

| Method/Dataset | METABRIC | Reuters | Caltech101-20 | VOC | Caltech5V-7 | RBGD | MNIST-USPS | CCV | MSRVC1 | Scene15 |
|---|---|---|---|---|---|---|---|---|---|---|
| ACC | | | | | | | | | | |
| DSMVC | 47.78 | 51.22 | 41.58 | 66.86 | **92.71** | 44.93 | 84.20 | 19.70 | 80.48 | 39.26 |
| CVCL | 52.48 | 55.41 | 35.43 | 39.82 | 89.14 | 33.21 | 99.68 | 26.27 | 93.33 | 42.72 |
| ICMVC | 33.96 | 39.18 | 27.41 | 38.80 | 64.36 | 34.92 | 99.42 | 22.04 | 77.14 | 42.36 |
| RMCNC | 34.58 | 44.63 | 43.17 | 42.49 | 58.36 | 34.37 | 83.00 | 23.90 | 35.24 | 41.32 |
| COPER | **55.28** | **58.96** | **58.26** | **67.76** | 88.36 | **53.21** | **99.94** | **29.13** | **93.81** | **44.68** |
| ARI | | | | | | | | | | |
| DSMVC | 22.88 | 26.72 | 35.51 | **58.94** | **85.10** | 29.01 | 76.86 | 6.59 | 64.05 | 23.48 |
| CVCL | **53. 92** | **38.83** | 25.96 | 27.03 | 77.33 | 17.59 | 99.68 | **29.68** | 84.99 | 26.10 |
| ICMVC | 12.24 | 16.65 | 18.00 | 26.12 | 44.42 | 17.37 | 98.72 | 6.23 | 55.88 | 26.47 |
| RMCNC | 11.90 | 14.06 | 30.85 | 21.61 | 39.89 | 16.46 | 66.41 | 9.98 | 12.62 | 25.46 |
| COPER | 33.01 | 30.59 | **51.55** | 58.28 | 76.26 | **36.68** | **99.87** | 12.88 | **86.74** | **26.40** |
| NMI | | | | | | | | | | |
| DSMVC | 30.07 | 33.55 | **62.34** | **69.10** | **87.11** | **41.63** | 83.54 | 17.43 | 68.43 | 41.93 |
| CVCL | 39.16 | **42.22** | 56.95 | 33.29 | 79.76 | 27.68 | 99.05 | 22.52 | 86.38 | **42.64** |
| ICMVC | 19.11 | 20.73 | 42.64 | 43.22 | 44.42 | 28.46 | 98.28 | 15.75 | 62.36 | 44.18 |
| RMCNC | 16.99 | 19.24 | 40.55 | 25.09 | 49.74 | 29.21 | 67.27 | 17.65 | 21.26 | 41.96 |
| COPER | **40.54** | 37.42 | 56.15 | 60.82 | 79.17 | 37.78 | **99.81** | **26.93** | **88.71** | 40.53 |

# I BROADER IMPACT

Our work presents a new deep learning-based solution for multi-view clustering, along with a theoretical foundation for its performance. This research can positively impact various domains by providing researchers and practitioners with a versatile data analysis tool for use across heterogeneous datasets, thereby facilitating advancements in knowledge discovery and decision-making processes. However, the proposed method has ethical implications and potential societal consequences that must be considered. It is crucial to pay attention to bias and fairness to prevent the amplification of biases across modalities. Transparency and explainability are essential to ensure user understanding and mitigate the perceived black-box nature of deep learning models.

# J COPER ALGORITHM

The COPER Algorithm processes multi-view data using deep canonically correlated encoders and clusters the data. It iterates through several epochs, performing different tasks based on the current epoch. The following is a detailed explanation of the algorithm:

---

**Algorithm** COPER

---

**Inputs:** Multi-View Data: $\mathcal{X} = \{\boldsymbol{X}^{(v)} \in \mathbb{R}^{d_v \times N}\}_{v=1}^{n_v}$, Deep Canonically Correlated Encoders: $\{\mathcal{F}^{(v)}\}_{v=1}^{n_v}$, Cluster Head: $\mathcal{G}$, Number of clusters $K$, Number of Epochs $N_{epochs}$, Epoch to Start Pseudo-Labeling and Permutations $N_{start}$.

**Outputs:** Cluster Assignments $\mathcal{Y}$ for each instance tuple $(\boldsymbol{x}_i^{(1)}, \boldsymbol{x}_i^{(2)}, ..., \boldsymbol{x}_i^{(n_v)})$, $i = 1, ..., N$

1: **for** epoch in 1 to $N_{epochs}$ **do**
2:     **for** each pair of views $v, w$ **do**
3:         Train encoders $\mathcal{F}^{(v)}$ and $\mathcal{F}^{(w)}$ with data $\boldsymbol{X}^{(v)}$ and $\boldsymbol{X}^{(w)}$ by minimizing $\mathcal{L}_{corr}$
4:     **end for**
5:     **if** epoch $> N_{start}$ **then**
6:         Predict cluster probabilities $\mathcal{G}(\sum_v w_v \boldsymbol{H}^{(v)}) = \boldsymbol{P}$
7:         Select top $B$ samples per cluster $\mathcal{T}_k = \{i \,|\, i \in \text{argtopk}(\boldsymbol{P}_{:,k}, B)\}$
8:         Filter samples in $\mathcal{T}_k$ using cosine similarity $\lambda$ and multi-view agreement
9:         Create view specific pseudo-labels $\mathcal{X}^{(v)} = \{(\boldsymbol{x}_i, \hat{\boldsymbol{y}}_i^{(v)})\}_{i \in \mathcal{T}}$
10:        **for** each view $v$ **do**
11:           Minimize $\mathcal{L}_{ce}$ with pseudo-labels for view $v$
12:        **end for**
13:        Create permuted multi-view data $\tilde{\mathfrak{X}} = \{\tilde{\mathcal{X}}^{(v)}\}_{v=1}^{n_v}$
14:        **for** each pair of views $v, w$ **do**
15:           Train encoders $\mathcal{F}^{(v)}$ and $\mathcal{F}^{(w)}$ with data $\tilde{\boldsymbol{X}}^{(v)}$ and $\tilde{\boldsymbol{X}}^{(w)}$ by minimizing $\mathcal{L}_{corr}$
16:        **end for**
17:     **end if**
18: **end for**

---

