# OpenReview forum: "COPER: Correlation-based Permutations for Multi-View Clustering"
_ICLR.cc/2025/Conference — ICLR 2025 Spotlight_

### Official Review · Reviewer_Yjr1 · 2024-10-31

**Soundness:** 3
**Presentation:** 3
**Contribution:** 3
**Rating:** 8
**Confidence:** 4

**Summary:**

This paper proposed an end-to-end MVC method that leveraged CCA-based correlation maximization and self-supervised pseudo-labels to learn multi-view representations and clusters jointly. In the proposed method, the key components are the Sample Selection and Label Refinement-Agreement of multi-view pseudo-labelling, which follows some technologies of semi-supervised learning and yields a simple unsupervised MVC architecture. Then, the authors present experimental and theoretical results to support the effectiveness of their method.

**Strengths:**

1. This paper presents a deep MVC method which has the advantages of simpleness and end-to-end. The method leverages a cross-entropy loss and a CCA-like maximization loss to train the deep model. It transfers the two-step training schedule in previous methods into step-by-step one, among which it conducts sample selection by high confidence and label correction by multi-view agreement.

2. The paper is well-written and easy to follow, which introduces theoretical proofs and visualization to support its method.

3. The illustration of cluster permutations is interesting, and it can enhance the embeddings learned by CCA (verified by ablation study).

**Weaknesses:**

I have the following concern and hope they are useful for improving this manuscript:

As for unsupervised clustering task, the robust model is needed when it processes different datasets in practical scenarios. However, we can observe that the proposed method is sensitive to model architecture settings (Table 7&9). For different datasets, the proposed method has different settings of model architecture and batch size. Moreover, in D.4 GRADUAL TRAINING, the training epochs in each step are determined by human. Since we have no labelled data to tune the model settings in practical applications, the proposed might have limited practical application value.

**Questions:**

1. Please see above weakness. It is encouraged to use a uniform model architecture to test the clustering performance of the proposed method on different datasets, for a fair comparison and availability.

2. It is encouraged to compare some latest self-supervised deep MVC approaches, e.g., On the effects of self-supervision and contrastive alignment in deep multi-view clustering [CVPR 2023], Investigating and Mitigating the Side Effects of Noisy Views for Self-Supervised Clustering Algorithms in Practical Multi-View Scenarios [CVPR 2024]...

---

> ### Author Response · Authors · 2024-11-20
> **Authors' response**
>
> We thank the reviewer for thorough and constructive feedback on our submission. We appreciate the acknowledgment of the strengths of our approach, particularly the simplicity and effectiveness of the end-to-end framework, the theoretical proofs and visualizations, and the innovative integration of multi-view pseudo-labeling techniques. Below, we provide detailed responses to the reviewer's comments.
>
> **Q1 Uniform Model Architecture + Weaknesses**
>
> We appreciate the feedback on uniform model architecture. To address this concern, we have conducted an experiment to demonstrate that the performance of our model is not too sensitive to batch size choices. Specifically, we trained our model with varying batch sizes in [128, 256,512,1024], with 5 random initializations for each batch size. Then, we compute the metrics ACC, ARI, and NMI for each setup and provide a box-plot chart in Appendix E.3. This experiment is an addition to the sensitivity analysis we provided for hyperparameter $\lambda$ in E.3.
>
> Variations in model architecture are influenced by the complexity of the dataset, such as feature dimensions, and align with established practices in multi-view clustering. Our benchmark methods, which have different configurations tailored to specific datasets, include CVCL [3], ICMVC [5], RMCNC [4], and MVCAN [2].
> Furthermore, although our gradual training strategy is straightforward, it can be automated using heuristic criteria once the loss converges to a predefined threshold. This approach reduces the need for manual adjustments. Other benchmark methods, such as CVCL [3] and RMCNC [4], also incorporate various gradual steps as part of their frameworks.
>
> Finally, in Appendix E2, we demonstrate how the Silhouette score, an unsupervised metric, could be used to tune model hyperparameters.
>
> **Q2 Compare with Latest Self-Supervised Approaches**
>
> We appreciate the Reviewer’s suggestion to compare our approach with the latest self-supervised multi-view clustering methods and have included a comparison to MVCAN [1] and L0-DCCA [7], OPMC [6].
> As shown in Tables 2 and 6, COPER consistently outperforms MVCAN [1] and L0-DCCA [7], OPMC [6] across most datasets at ACC, ARI, and NMI, demonstrating its robustness and adaptability across diverse clustering scenarios. Although MVCAN achieves competitive performance in specific cases, such as Caltech101-20 for NMI, its results are more variable and generally lower than COPER, particularly on complex or noisy datasets like RBGD and VOC.
>
> We tried to use the DeepMVC [2] Git repository but faced considerable challenges in reproducing the environment. We were unable to establish a working environment, and the package dependencies led to complex compatibility issues that led to initialization errors. We will work toward solving these issues by reaching out to the owners of this package via Github.
>
> **References**:
>
> [1] Investigating and Mitigating the Side Effects of Noisy Views for Self-Supervised Clustering Algorithms in Practical Multi-View Scenarios (CVPR 2024)
>
> [2] On the Effects of Self-Supervision and Contrastive Alignment in Deep Multi-View Clustering" (CVPR 2023)
>
> [3] Deep Multiview Clustering by Contrasting Cluster Assignments" is accepted by ICCV 2023
>
> [4] Sun, Y., Qin, Y., Li, Y., Peng, D., Peng, X., & Hu, P. (2024). Robust multi-view clustering with noisy correspondence. IEEE Transactions on Knowledge and Data Engineering.
>
> [5] Chao, G., Jiang, Y., & Chu, D. (2024, March). Incomplete contrastive multi-view clustering with high-confidence guiding. In Proceedings of the AAAI Conference on Artificial Intelligence
>
> [6] Liu J, et al. One-pass Multi-view Clustering for Large-scale Data, ICCV 2021.
>
> [7] Lindenbaum, O., Salhov, M., Averbuch, A., & Kluger, Y. (2021). L0-sparse canonical correlation analysis. In International Conference on Learning Representations.

---

> > ### Author Response · Authors · 2024-11-25
> >
> > Dear reviewer, we appreciate the time and effort you have dedicated to reviewing our paper. As the discussion period is limited, we kindly request that the reviewer evaluate the new information provided in the rebuttal. We are eager to improve our paper and resolve all the concerns raised by the reviewer. If there are any remaining concerns that have not been addressed, we would be happy to provide further explanations.

---

> > > ### Comment · Reviewer_Yjr1 · 2024-11-27
> > >
> > > Thanks for authors' rebuttal and it almost addresses all my concerns. Perhaps because of the different experimental environment, the results of many methods on some datasets are different from the original paper (e.g., CVCL on MSRC-v1 and  Scene15, MVCAN on MNIST-USPS/DIGIT and RGB-D, ......). Although it is nearly impossible to achieve the exact same experimental conditions for all methods, we want to ensure that the experimental setup is as consistent as possible. Anyway, the most important point of the paper is not the experimental ACC but the innovation, so I tend to raise my score to ''accept''.

---

### Official Review · Reviewer_bbo9 · 2024-10-31

**Soundness:** 3
**Presentation:** 2
**Contribution:** 3
**Rating:** 8
**Confidence:** 5

**Summary:**

In literature, most of the current multi-view clustering methods are limited to specific domains or rely on a sub-optimal and computationally intensive two-stage process of representation learning and clustering. To address this issue, the authors propose an end-to-end deep learning-based multi-view clustering framework which is validated to be effective in experiments.

**Strengths:**

1. The theory of LDA is analyzed.
2. The proposed method is validated to be effective in experiments.

**Weaknesses:**

1. The authors say that most of existing multi-view clustering methods are composed of two-stage process of representation learning and clustering, therefore they propose an end-to-end method. However, the authors also claim that a few end-to-end methods are proposed in literature. So, the motivation should be further clarified.

2. The paper [1] is an classical end-to-end multi-view clustering method and should be compared or discussed.

3. The parameter study should be included.

[1] Liu J, et al. One-pass Multi-view Clustering for Large-scale Data, ICCV 2021.

**Questions:**

Please see Weakness

---

> ### Author Response · Authors · 2024-11-20
> **Authors' response**
>
> We thank the reviewer for their thoughtful and constructive feedback and for acknowledging the strengths of our work, particularly the effectiveness of our end-to-end multi-view clustering framework and the validation of our theoretical contributions through extensive experiments. We address the reviewer's specific comments and suggestions in detail below.
>
> **W1 Motivation**
>
> We thank the reviewer for their valuable feedback. To clarify, our motivation stems from the limitations of existing multi-view clustering (MVC) methods, particularly those that adopt a two-stage process where representation learning is followed by clustering. This two-stage process can be suboptimal, as the representations learned are not directly optimized for clustering, often resulting in subpar performance. While we acknowledge that a few end-to-end methods have been proposed, they are often constrained by limited adaptability to various data types or require specific assumptions that restrict their generalizability.
> Our work proposes an end-to-end approach that is distinct in leveraging self-supervised learning to improve canonical correlation analysis (CCA)-based clustering. Specifically, we incorporate a novel self-supervision scheme into an end-to-end clustering procedure, which can enhance any CCA-based method with better representations tailored for clustering. Unlike existing end-to-end approaches, our method introduces within-cluster permutations and multi-view pseudo-labeling, which allow for improved cluster separation and representation alignment, thereby addressing the inherent limitations of prior approaches. This motivation is grounded in the understanding that direct optimization of the representations for clustering tasks can lead to better generalization and more effective model performance across different data types, as evidenced by our extensive empirical results.
>
> **W2 Additional baselines**
>
> Thank you for suggesting this end-to-end approach. In response, we introduced an evaluation of OPMC [1], L0-CCA [2], and MVCAN [3] in the revised version (see Tables 2 and 6). OPMC performs well against other baselines and outperforms COPER in 2 out of 10 datasets regarding accuracy (ACC). However, COPER still surpasses OPMC, L0-CCA, and MVCAN in most metrics and across most datasets.
>
> **W3 Parameter Study**
>
> Thank you for your suggestion. We provide a sensitivity analysis for $\lambda$ value in Appendix E.3 and extend it with an additional analysis of the batch size hyperparameter. We train our model on the MSRCv1 dataset with batch sizes [128, 256, 512, 1024] five times for each batch size, each time with a different random initialization seed. Then, we measure the metrics ACC, ARI, and NMI and put the results on the box-plot chart (E.2 Figure 6).
>
> **References**:
>
> [1] Liu J, et al. One-pass Multi-view Clustering for Large-scale Data, ICCV 2021.
>
> [2] Lindenbaum, O., Salhov, M., Averbuch, A., & Kluger, Y. (2021). L0-sparse canonical correlation analysis. In International Conference on Learning Representations.
>
> [3] Investigating and Mitigating the Side Effects of Noisy Views for Self-Supervised Clustering Algorithms in Practical Multi-View Scenarios (CVPR 2024)

---

> > ### Author Response · Authors · 2024-11-25
> >
> > Dear reviewer, we appreciate the time and effort you have dedicated to reviewing our paper. As the discussion period is limited, we kindly request that the reviewer evaluate the new information provided in the rebuttal. We are eager to improve our paper and resolve all the concerns raised by the reviewer. If there are any remaining concerns that have not been addressed, we would be happy to provide further explanations.

---

> > > ### Comment · Reviewer_bbo9 · 2024-11-27
> > > **Thanks for the response**
> > >
> > > Thanks for the responses. They have addressed my concrens and I would like to raise my rating to "accept".

---

### Official Review · Reviewer_gjrx · 2024-11-04

**Soundness:** 3
**Presentation:** 3
**Contribution:** 3
**Rating:** 8
**Confidence:** 4

**Summary:**

This paper proposes a deep learning model for multi-view clustering framework, namely, COPER. The proposed model integrates clustering and representation tasks into an end-to-end framework, eliminating the need for a separate clustering step. Extensive experimental evaluation across various benchmark datasets validates the efficiency of the proposed algorithm.

**Strengths:**

1.	This paper is well organized, and the proposed methodology is enlightening.
2.	The motivation behind the paper is clear, and the theoretical analysis is complete.
3.	The comparison experiments are comprehensive, encompassing datasets of varying sizes and multiple types of baseline methods.

**Weaknesses:**

1.	Unlike general methods, the proposed approach generates pseudo-labels for each view to enable self-supervised learning. However, in multi-view clustering, aligning the labels across views can pose challenges that may impact subsequent self-supervised learning.
2.	Although the paper includes theoretical analysis, the proposed method offers limited innovation. The correlation maximization loss has already been proposed, and generating pseudo-labels by estimating the probability matrix is also a common approach.

**Questions:**

As mentioned above.

---

> ### Author Response · Authors · 2024-11-20
> **Authors' response**
>
> We thank the reviewer for their thoughtful and detailed feedback. We are grateful for the acknowledgment of the strengths of our work, including the clarity of our motivation, the rigor of our theoretical analysis, the breadth of our comparative experiments, and the well-structured presentation of the paper. Below, we respond to all comments raised by the reviewer.
>
>
> **W1 Alignment Pseudolabels**
>
> Thank you for your feedback. As you indicated, label alignment across views in multi-view clustering presents challenges that could potentially limit the effectiveness of self-supervised learning methods.
>
> To address potential issues with pseudo-label alignment, our framework uses a multi-view agreement mechanism to refine pseudo-labels and enhance their reliability. This step (elaborated in Section 4.4.2) reduces errors by focusing on consistent labels across views.
> Even when some pseudo-labels are incorrect, our learned representation still improves clustering. As shown in Fig. 3(i)(a), despite some level of falsely annotated pseudo-labels (24% error), the Adjusted Rand Index still improves (purple line), demonstrating the robustness of our approach. Additionally, our theoretical analysis in Sec. 4.3 provides bounds that further support the method's resilience to pseudo-label errors.
>
> **W2 Technical innovation of our work**
>
> While specific individual components like correlation maximization and pseudo-labeling have appeared in prior work, COPER introduces the integration of these components along with several additional innovations. First, we present a novel framework for multi-view pseudo labeling, which includes multi-view agreement. This agreement is essential for facilitating within-cluster permutations in the subsequent steps. Additionally, although the relationship between Canonical Correlation Analysis (CCA) and Linear Discriminant Analysis (LDA) has been previously examined, it has not been applied to multi-view clustering frameworks. In addition to our proposed method for enabling within-cluster permutations, we leverage the relationship between CCA and LDA by providing an approximation analysis and deriving new error bounds for our learned representations, utilizing matrix perturbation theory. This supports the effectiveness of our approach.

---

> > ### Comment · Reviewer_gjrx · 2024-12-03
> >
> > Thanks for your response. My concerns have been resolved, so I recommend accepting this manuscript.

---

### Official Review · Reviewer_aFs1 · 2024-11-04

**Soundness:** 3
**Presentation:** 3
**Contribution:** 3
**Rating:** 5
**Confidence:** 4

**Summary:**

The proposed approach involves generating meaningful fused representations using a novel permutation-based canonical correlation objective. Cluster assignments are learned by identifying consistent pseudo-labels across multiple views.

**Strengths:**

The proposed approach involves a novel permutation-based canonical correlation objective. Simulanteously, the authors provide a theoretical analysis showing how the learned embeddings approximate those obtained by supervised linear discriminant analysis (LDA).

**Weaknesses:**

1）In Line 41, multi-view clustering holds immense potential in various applications, however, the methods mentioned are not updated to recent literature. Please update these references to ensure your work is current.

2）The selected comparison methods are not enough. It is recommended to add some comparison methods, otherwise this may have a negative impact on the reliability of the experimental results.

3）Some of the selected comparison methods are for incomplete multi-view data, and some are for noise correspondence. These methods have special properties and are not recommended as comparison methods.

4）Considering that the proposed method is derived from the CCA objective, it is recommended to classify the compared methods into CCA-derived methods and other non-CCA-derived methods. This can directly demonstrate the effectiveness of the new elements introduced by the proposed method compared to previous similar frameworks and its competitiveness compared to other MVC paradigms.

5) Check for all possible errors in the statement, e.g. the missing serial number for Figure in Line 244,“using using within-cluster permutations”in Line 292.

**Questions:**

See above.

---

> ### Author Response · Authors · 2024-11-20
> **Authors' response**
>
> We thank the reviewer for the valuable feedback. We appreciate the recognition of our work's strengths, including the novel permutation-based canonical correlation objective and the theoretical analysis showing how the learned embeddings approximate those obtained by supervised linear discriminant analysis (LDA). Below, we respond to all comments and outline our modifications to address the raised issues.
>
> **W1 Updated multi-view applications**
>
> We have updated the text to include motivating examples of multi-view clustering applied to more recent use cases. The revised text  (page 1, lines 41-44) is:
> "This approach has great potential in applications like communication systems content delivery [1], community detection in social networks [2,3], cancer subtype identification in bioinformatics [4], and personalized genetic analysis through multi-modal clustering frameworks [5,6]."
> Updated references:
>
> **References**:
>
> [1] Miguel Angel Vázquez and Ana I Pérez-Neira, "Multigraph Spectral Clustering for Joint Content Delivery and Scheduling in Beam-Free Satellite Communications," ICASSP 2020.
>
> [2] Zhao et al., "Multi-view Tensor Graph Neural Networks Through Reinforced Aggregation," IEEE TKDE 2022.
>
> [3] Shi, X., Liang, C., and Wang, H., "Multiview Robust Graph-based Clustering for Cancer Subtype Identification," IEEE/ACM Transactions on Computational Biology and Bioinformatics, 2023.
>
> [4] Wang, B., et al., "Multi-dimensional Data Integration for Personalized Analysis Using Random Walks," BMC Bioinformatics, 2023.
>
> [5] Li et al., "netMUG: A Network-guided Multi-view Clustering Workflow for Dissecting Genetic and Facial Heterogeneity," Frontiers in Genetics, 2023.
>
> [6] Wen et al., "Deep Multi-view Clustering with Contrastive Learning for Omics Data," Bioinformatics, 2024.

---

> ### Author Response · Authors · 2024-11-20
> **Authors' response cont.**
>
> **W2/W3/W4 Baselines**
>
> We appreciate the reviewer’s feedback regarding our choice of baselines and the suggestions for expanding and categorizing the methods. ICMVC [7] and RMCNC [8] were chosen since they were recently published and because of their demonstrated effectiveness and relevance in various multi-view clustering scenarios beyond their specialized design focuses.
> ICMVC [7] was originally designed to manage incomplete multi-view data. However, it includes robust components such as instance-level attention fusion and contrastive learning for view alignment, which remain highly effective even when all views are fully observed. Additionally, its weight-sharing pseudo-classifier allows for end-to-end representation and clustering, making it a strong competitor in our context. The evaluation results presented in Table 1 of the ICMVC paper [7] show excellent performance on standard datasets without any missing views, further supporting our choice to include it as a baseline.
> Similarly, RMCNC's core architecture [8] is designed to handle noisy correspondences effectively. It incorporates components such as a noise-tolerant contrastive learning framework and a unified probability computation strategy, which enhance its clustering performance across various data conditions. The evaluation results (refer to Table 2 in the original paper) demonstrate strong performance on noise-free datasets, highlighting its suitability as a comparison method. Since real-world multi-view datasets frequently contain noisy correspondences, RMCNC is a practical choice for our evaluation.
>
> Following the reviewer's suggestion, we have expanded our experiments to include additional baselines that cover a broader spectrum of multi-view clustering techniques [9, 10, 11]. As suggested, we have categorized our baselines into CCA-derived and non-CCA-derived methods (see Tables 2 and 6). This categorization highlights the distinct contributions of our method’s novel permutation-based CCA objective, clearly assessing its advantages over similar frameworks and alternative multi-view clustering paradigms.
>
>
>
> Our systematic experimental evaluation involves ten diverse datasets (2–6 views each). We benchmarked against ten baselines, conducted an ablation of four studies, and assessed scalability. Our method demonstrates superior performance, surpassing eight state-of-the-art deep models with up to 7% improvement in accuracy and consistently higher ARI and NMI scores across datasets. Robustness is ensured by averaging results over ten runs and reporting standard deviations, providing a comprehensive and reliable performance assessment.
>
> **References**:
>
> [7]. Chao, G., Jiang, Y., & Chu, D. (2024, March). Incomplete contrastive multi-view clustering with high-confidence guiding. In Proceedings of the AAAI Conference on Artificial Intelligence
>
> [8]. Sun, Y., Qin, Y., Li, Y., Peng, D., Peng, X., & Hu, P. (2024). Robust multi-view clustering with noisy correspondence. IEEE Transactions on Knowledge and Data Engineering.
>
> [9] Liu J, et al. One-pass Multi-view Clustering for Large-scale Data, ICCV 2021.
>
> [10] Investigating and Mitigating the Side Effects of Noisy Views for Self-Supervised Clustering Algorithms in Practical Multi-View Scenarios (CVPR 2024)
>
> [11] Lindenbaum, O., Salhov, M., Averbuch, A., & Kluger, Y. (2021). L0-sparse canonical correlation analysis. In International Conference on Learning Representations.
>
>
> **W5 Minor comments and corrections**
>
> We appreciate your feedback regarding the necessary changes. We have reviewed the paper again and made adjustments based on your suggestions.

---

> > ### Author Response · Authors · 2024-11-25
> >
> > Dear reviewer, we appreciate the time and effort you have dedicated to reviewing our paper. As the discussion period is limited, we kindly request that the reviewer evaluate the new information provided in the rebuttal. We are eager to improve our paper and resolve all the concerns raised by the reviewer. If there are any remaining concerns that have not been addressed, we would be happy to provide further explanations.

---

> > > ### Author Response · Authors · 2024-11-29
> > > **End of discussion period**
> > >
> > > As the discussion period is nearing its end, we would like to check if there are any remaining concerns regarding our paper. In response to the reviewer’s feedback, we have implemented an evaluation of three additional methods and reorganized our results table to separate CCA-based and non-CCA-based approaches.
> > >
> > > We hope these revisions address all concerns raised, and we would be grateful for any further feedback before the discussion concludes.

---

### Author Response · Authors · 2024-11-20
**General Response**

We thank the reviewers for their detailed feedback and insightful comments on our work. We appreciate their recognition of our permutation-based canonical correlation objective and its connection to supervised Linear Discriminant Analysis (LDA) projections. We are also grateful for their acknowledgment of our end-to-end framework, which effectively integrates clustering and representation learning to address traditional multi-view clustering challenges. Additionally, we value their positive remarks on the clarity and organization of our work, supported by our theoretical analysis and experimental validations.

In response to the reviews, we have carefully addressed the concerns raised to significantly enhance the manuscript. We have updated the background section based on the reviewers' feedback. Additionally, we've expanded the experimental comparisons to include more baseline methods, and we have categorized our approaches into CCA-derived and non-CCA-derived methods. We demonstrated a unified experimental setup with consistent architectural and parameter settings across datasets to showcase practical applicability and strengthen the robustness of our findings.

Below, we have provided a detailed response to each reviewer’s comments. We encourage the reviewers to refer to the revised manuscript (and Appendix), where we have highlighted all updates in color. We hope that these revisions satisfactorily address the reviewers’ questions and concerns. We greatly appreciate the feedback and are open to any further clarification or discussion.

---

### Meta-Review · Area_Chair_9jqp · 2024-12-19

**Metareview:**

This paper introduces an end-to-end MVC approach that integrates CCA-based correlation maximization with self-supervised pseudo-labeling, achieving joint multi-view representation learning and clustering. Both experimental and theoretical analyses are provided to demonstrate the efficacy of the proposed method. Based on the positive feedback from three reviewers, I decide to accept the paper. However, as pointed out by Reviewer Yjr1, the authors need to explicitly discuss the sensitivity of the proposed method to hyperparameters and network architectures.

**Additional Comments On Reviewer Discussion:**

After the authors' rebuttal, three reviewers raised their scores to accept. Reviewer aFs1 did not respond to the authors, but I think the authors' response can address the concerns. Thus I would like to accept this paper.

---

### Decision · Program_Chairs · 2025-01-22

Accept (Spotlight)